# On–off conduction photoswitching in modelled spiropyran-based metal-organic frameworks

Mersad Mostaghimi[1,3], Helmy Pacheco Hernandez[1,3], Yunzhe Jiang[2], Wolfgang Wenzel [1], Lars Heinke [2✉] & Mariana Kozlowska [1✉]

Materials with photoswitchable electronic properties and conductance values that can be reversibly changed over many orders of magnitude are highly desirable. Metal-organic framework (MOF) films functionalized with photoresponsive spiropyran molecules demonstrated the general possibility to switch the conduction by light with potentially large on-off-ratios. However, the fabrication of MOF materials in a trial-and-error approach is cumbersome and would benefit significantly from in silico molecular design. Based on the previous proof-of-principle investigation, here, we design photoswitchable MOFs which incorporate spiropyran photoswitches at controlled positions with defined intermolecular distances and orientations. Using multiscale modelling and automated workflow protocols, four MOF candidates are characterized and their potential for photoswitching the conductivity is explored. Using ab initio calculations of the electronic coupling between the molecules in the MOF, we show that lattice distances and vibrational flexibility tremendously modulate the possible conduction photoswitching between spiropyran- and merocyanine-based MOFs upon light absorption, resulting in average on-off ratios higher than 530 and 4200 for p- and n-conduction switching, respectively. Further functionalization of the photoswitches with electron-donating/-withdrawing groups is demonstrated to shift the energy levels of the frontier orbitals, permitting a guided design of new spiropyran-based photoswitches towards controlled modification between electron and hole conduction in a MOF.

[1] Institute of Nanotechnology (INT), Karlsruhe Institute of Technology (KIT), Kaiserstraße 12, 76131 Karlsruhe, Germany. [2] Institute of Functional Interfaces (IFG), Karlsruhe Institute of Technology (KIT), Kaiserstraße 12, 76131 Karlsruhe, Germany. [3] These authors contributed equally: Mersad Mostaghimi, Helmy Pacheco Hernandez. ✉email: lars.heinke@kit.edu; mariana.kozlowska@kit.edu

Remote-controllable materials, also referred to as smart materials, are functional materials whose properties can be switched by external stimuli without direct contact. Light is a particularly simple, handy and (usually) non-invasive signal, which enables a very fast response of the smart material with high spatial resolution. Therefore, photoswitchable materials, which incorporate photoswitchable or photochromic molecules, attract a lot of attention[1,2]. Photoswitchable molecules undergo isomerization when irradiated by light of a certain wavelength and reversibly isomerize back to the original state by irradiation with a different wavelength or by thermal relaxation. Apart from photoswitching the color of the material, switching of the electronic properties is especially interesting enabling applications in various fields, ranging from optical switches over electro-optical devices to optical data storage and optical computing.

Spiropyran (SP) is a photochromic molecule, which can undergo light-induced isomerization from the closed SP-form to the open zwitterionic merocyanine (MC) form by UV light and vice versa by visible light or thermal relaxation. The SP-MC isomerization is correlated with a switching of the electronic molecular properties[3,4]. It is accompanied by a length change of the molecule, for nitro-functionalized SP from 12.2 Å to 14.0 Å, see Fig. 1. It is particularly interesting that SP-MC switching induces significant changes of the electronic structure resulting in significant shifts of the frontier orbitals, HOMO (Highest Occupied Molecular Orbital) and LUMO (Lowest Unoccupied Molecular Orbital), with respect to the energy level. Upon SP-to-MC switching, the π-electron density localization decreases and the extension of the frontier orbitals changes: While the HOMO and LUMO in the SP form are localized at the indoline and the chromene moiety, respectively, the HOMO and LUMO of MC are spread over the entire molecule[5]. It was demonstrated that the SP-to-MC isomerization results in an increase of the conductivity of single spiropyran molecules[6] and self-assembled spiropyran monolayers[7,8]. It was shown that the change in the conduction of self-assembled spiropyran monolayers is a result of tunneling effects between the orbitals[7].

The switching of pure SP is barely possible, since it is a crystalline solid at room temperature where the light-induced SP-to-MC isomerization is sterically hindered. At the same time, switching of spiropyran in its solvated forms limits future device implementation. Therefore, a major challenge is to explore the conductance switching based on molecular photoisomerization of SP within three-dimensional materials. Due to the lack of

conformational freedom in polymers, the isomerization of incorporated molecules like spiropyran, whose isomerization involves large geometrical changes, is typically hindered and is only possible for very small concentrations of photoswitchable monomers[9]. Other photochromic molecules, suitable for the remote control of the electronic properties, are azobenzenes[10] and diarylethenes[11]. When incorporated in conducting polymers, the conduction could be switched by light between a highly and a barely conducting state[12,13]. This effect is caused by photoswitching the HOMO and LUMO levels, which act as scattering centers for the conduction along the polymer orbitals[13] or change the charge injection[12]. While polymers impressively demonstrate the potential of photoswitchable materials, oriented and anisotropic conduction and its switching have not been demonstrated in these systems. A further major drawback of polymers is the lack of crystallinity, resulting in a molecular environment that is difficult to control. This hinders the design of polymers in photoswitching applications and complicates theoretical predictions. As a result, research of polymeric photoswitchable materials relies to a large degree on trial and error. Moreover, the relatively low density of photoswitches in the material additionally limits their performance. Thus, although impressive effects could be demonstrated in these experiments, the preparation of solid photoswitchable materials is still the bottleneck for applications and design rules are still missing. A precise architecture of the material, which incorporates the smart, photoresponsive components at well-defined positions and orientations, is required but has not yet been realized.

One approach, which overcomes these limitations and enables the upscaling of the functionality of a single molecule to yield functional solid materials or thin films of macroscopic dimensions, is offered by integration of the photoswitches into metal-organic frameworks, MOFs[14,15]. These highly porous, crystalline solid materials consist of metal (or metal-oxo) nodes connected by organic linker molecules. By combining various metal ions and organic linkers, a huge variety of MOF has been realized and more than 100,000 different MOF structures have been synthesized so far. MOF properties can be designed and tuned by choosing the appropriate components[16–18]. A major advantage of MOFs, in comparison to polymers, is the higher density of photoswitches and their ordered assembly: typically, one or more photoswitchable molecules can be incorporated in one unit cell, corresponding to roughly one photoswitch per cubic nanometer. In addition, the crystalline MOF structure facilitates quantum-mechanical calculations, e.g. by means of density-functional theory (DFT), enabling a detailed understanding of the fundamental processes as well as development of structure-property predictions[5,19–22]. Finally, due to their periodic, highly ordered structure and pronounced thermal stability, they can be characterized with high resolution.

The electronic properties of MOFs are intensively investigated, mainly with the focus on advanced semiconductors, energy storage and energy harvesting applications[23–25]. The conductivity of most MOFs, which typically are wide-bandgap semi-conductors, is rather low. Modeling shows that, as a result of the charge localization, large distance between the molecular units in the MOF and their vibrational flexibility[17,26], the electronic coupling in the crystalline framework is generally small. This often leads to low charge transfer (CT) rates in the material and small conductivities. Conventional charge hopping[5,27], transport via superexchange mechanism[28] and charge transport at the borderline between the localized hopping and delocalized charge transport[26,29] were shown as common charge transport mechanisms in MOFs. The conductivity of MOFs can be increased via tuning the structural parameters of MOF and virtual design of new MOF candidates with adequate electronic

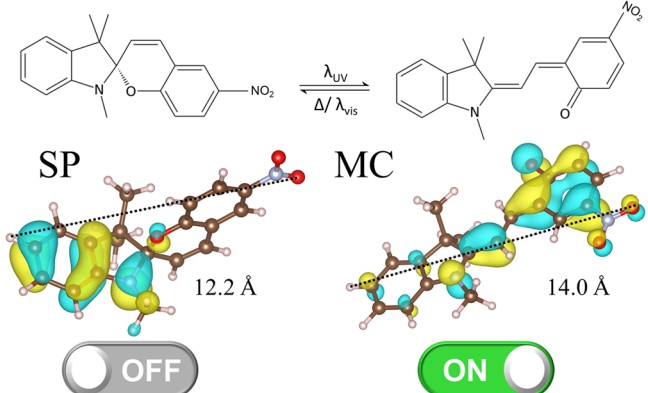

**Fig. 1 Photoswitching of spiropyran.** The SP-MC isomerization with the visualization of the HOMO. The SP-to-MC isomerization typically occurs by UV light ($\lambda_{UV}$), while MC-to-SP occurs by thermal relaxation or irradiation with visible light ($\lambda_{vis}$). The C atoms are shown in brown, O in red, N in light blue, and H in white.

structures[17,24,30]. As an example, it can be increased by realizing structures which enable efficient charge transfer through bonds[31] or through space[28], e.g. by π-stacking[32]. In this way, MOFs with conductivities in the range of organic semi-conductors were realized. In other approaches, the conductivity was significantly increased by incorporating guest molecules like TCNQ[24], fullerene[27,33], or ferrocene[34] in the MOF pores. Tailoring the MOF conductivity via design of new linkers with tuned HOMO/LUMO levels, position in the MOF and their improved electron coupling was also postulated[17,35]. For precise and reproducible conduction measurements, thin films are more suitable than MOF pellets, where the resistance of grain boundaries may dominate[24,25]. In addition, in contrast to a thin film where the entire material of can be illuminated (and switched by light), for materials in the form of powders or pellets, only the outer layer is illuminated due to the short light penetration depth in such materials with a high density of photochromic molecules[36,37].

Since their first realization about 10 years ago, many MOFs with photoswitchable moieties have been investigated for remote-controlling various properties, see these recent review articles[36,38–40]. The most popular photoswitchable components in MOFs are azobenzenes, but diarylethenes were also widely used. In most applications, guest-host interactions are controlled by the photoisomerization. This results, for example, in remote-controlled switching of the adsorption capacity of guest molecules, such as $CO_2$[20,41,42] or methane[43], the controlled release of the guest molecules[44,45], the switching of the membrane separation factor[46,47] or of the proton-conduction[21]. Furthermore, the catalytic activity[48], the luminescence[49] and the color[50] of the MOF material have been controlled by organic photoswitches. Only a small number of publications address MOFs with photoswitchable spiropyran so far, where the spiropyran molecules are incorporated by embedment in the pores[5,51–53] or by attachment to the framework[54–57]. The photoswitching of the conductive and electronic properties of spiropyran-containing MOFs has been investigated by refs. [5,56]. In the first study, spiropyran was embedded in a MOF film, which has shown a reversible switching of the electronic conductance by one order of magnitude[5]. However, for successful device implementation an on-off-conductance ratio of several orders of magnitude is required, with a minute (leakage) off current and an ideally loss-free on current. A detailed analysis of the switching process indicated the possibility to tremendously increase the on-off ratio, if the spiropyran moieties have a defined intermolecular distance instead of the random distribution. In the second study, spiropyran-functionalized MOF linkers were used, enabling the switching effect of the conduction of only 20%[56]. This might be caused by the fact that MOF-pellets were used, where only the outer surface was illuminated and photoswitched (as a result of the strong molecular absorption) resulting in a very small amount of merocyanine, i.e. in a low photoconversion in the photo-stationary state.

In this article, we utilize multiscale modeling, comprising DFT calculations and molecular dynamics (MD) simulations, to predict electronic and on-off photoswitching properties of four spiropyran-based MOFs. The switchable SP⇌MC units are incorporated as layer linkers in the pillared-layer MOFs with the copper paddlewheel metal nodes. Several pillar linkers of different lengths are used to modify the on-off conduction switching and to reveal its dependence as a function of the MOF interlayer distance. Electron-donating functional groups are attached to the plain SP linker, altering the energy and the occupation of the frontier orbitals, impacting the electronic coupling between the linkers in the MOF. Significant vibrational flexibility of the layer linkers at room temperature, especially of the MC isomer, is found. This causes additional intermolecular interactions in the MOF, permitting an increase of two orders of magnitude in the electronic coupling of the MOF components, enabling a tremendous increase in the on-off conduction switching. The functionalization of the initial SP linker with electron-donating and electron-withdrawing functional groups alters the energy and occupation of the frontier orbitals, impacting the electronic coupling between the linkers in the MOF.

## Results and discussion
In total, eight Cu-paddlewheel pillared-layer MOF structures with two main types of layer linkers (marked in red in Fig. 2), i.e. linear dicarboxylates with spiropyran side groups and its isomer merocyanine, and two passive pillar linkers (marked in pink in Fig. 2), i.e. 1,4-diazabicyclo[2.2.2]octane (dabco) and bipyridine (bipy), were investigated. For the dabco-pillared MOFs, the layer linkers were additionally modified by OH electron-donating and CHO electron-withdrawing functional groups introduced as $R_1$. We name all MOFs in the following convention: $Cu_2(L)_2(P)$, where L and P stand for the layer and the pillar linker, respectively. In the following, we report results for $Cu_2(SP)_2(dabco)$, $Cu_2(SP)_2(bipy)$, $Cu_2(MC)_2(dabco)$ and $Cu_2(MC)_2(bipy)$ for spiropyran- and merocyanine-based linkers, respectively, depicted in Fig. 2, and $Cu_2(SP-OH)_2(dabco)$, $Cu_2(SP-CHO)_2(dabco)$, $Cu_2(MC-OH)_2(dabco)$, $Cu_2(MC-CHO)_2(dabco)$ for the MOFs with modified linkers. In addition, the electronic properties of several other linkers with further $R_1$, $R_2$ and $R_3$ functionalization (see Fig. 2) were calculated.

The linkers in the MOF can be (photo-)switched between its SP and MC form, as depicted in Fig. 1. In our previous study on spiropyran embedded in the UiO-67 MOF pores[5], a switching yield of ~70% MC was achieved upon 365 nm UV light irradiation. In general, the isomerization can occur near-quantitative in both directions[3]. In the present case, we assume (ideal) 100% conversion between the photo-isomers, in order to understand possible on-off conduction ratios as the function of the intersheet distances and of chemical modifications of linkers in the simulated MOF models. Therefore, MOFs with either spiropyran- or merocyanine-based side chains are considered. Our focus is on the ratio of the electronic coupling between SP and MC MOFs that is directly connected to the change in the conduction of the MOFs in both forms.

The structures of all MOFs were calculated using an automated workflow, reported previously[17] (Fig. S1). It comprises several software packages, performing calculations on different level of theory, and tools for an automated MOF model generation, (pre)-optimization and simulation. Detailed description of the work-flow is explained in the Supplementary Information (SI). The starting position of all layer linkers in the MOF was selected in the way that the side chains point in the same direction, as depicted in Figs. S3 and S4. The opposite location of the side chains does not permit the formation of the merocyanine-based MOFs, because the intermolecular distances between the neighboring linkers are too small. If the opposite location of the spiropyran side chain happens during MOF fabrication, it will limit the photoconversion to the merocyanine form, impacting the properties of the material. We exclude such effects in the present study, therefore, the calculated on-off photoswitching ratio in the present study should be considered as the highest possible value for the type of the investigated MOF. $Cu_2(SP)_2(dabco)$ and $Cu_2(MC)_2(dabco)$ were initially optimized using plane-wave DFT and simulated at room temperature using classical MD simulations. All other MOFs were pre-optimized with DFT and simulated with MD using the same computational setup. Computational details are described in the Methods section and explained in more detail in SI.

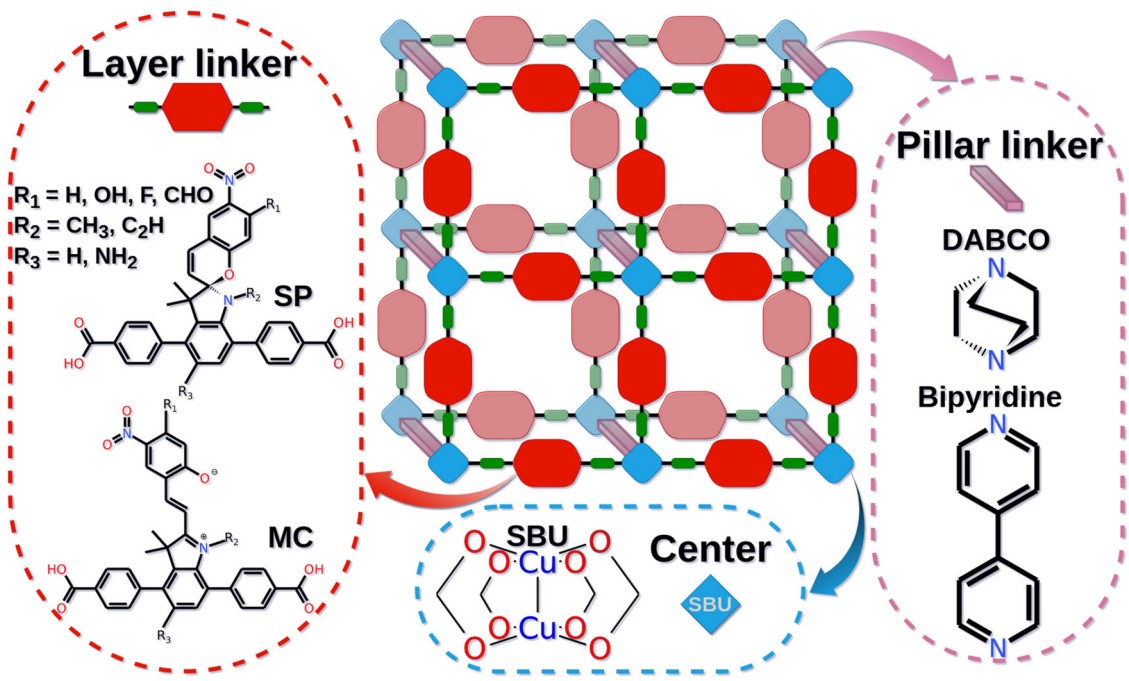

**Fig. 2 Schematic representation of investigated pillared-layer metal-organic frameworks.** To show the change of electronic properties of MOFs upon photoisomerization, both SP-based and their MC-based MOF counterparts were calculated. Several layer linker functionalizations ($R_1$, $R_2$ and $R_3$) were considered. Dabco and bipyridine were used as pillar linkers, while Cu-based paddlewheel node were used as secondary building unit (SBU). The MOF analysis was performed for the SP and MC linkers and linkers with $R_1 = H$, OH and CHO.

Significant differences in the electronic coupling between spiropyran and merocyanine molecules were previously reported to occur at larger distances[5]. In the present study, we focus on the change of the electronic coupling (i.e. transfer integral) between the linkers in the MOF upon 100% photo-isomerization, which indicates the degree of possible on-off conduction photoswitching. Electronic coupling is the microscopic property, describing the electronic interaction between two molecules or fragments, that leads to the transfer or sharing of electrons. It is connected to the probability of electron transfer between two molecules; therefore, it is often used in CT theories, e.g. in the Marcus theory of hopping transport[58]. Here, the higher the electronic coupling between the molecules was demonstrated to result in faster CT rate and higher charge carrier mobilities according to the following equation:

$$k_{CT} = \frac{2\pi}{\hbar} \left| J_{if} \right|^2 \sqrt{\frac{1}{4\pi\lambda k_B T}} \exp\left( \frac{-\left( \Delta E_{if} + \lambda \right)^2}{4\lambda k_B T} \right), \quad (1)$$

where $J_{if}$ corresponds to the electronic coupling (transfer integral) between initial and final molecular states in CT, i.e. $i$ and $f$, respectively. $\lambda$ is the reorganization energy, connected to the change in the equilibrium geometry of both acceptor ($f$) and donor ($i$) upon change of the charged state, $k_B$ is the Boltzmann constant, $\Delta E_{if}$ is the energy difference between frontier molecular orbitals involved in CT, and $T$ is temperature.

However, Marcus-type hopping transport represents only the localized CT transport, while other types of CT may also occur[59], e.g. band-like transport or CT on the border between delocalized and localized charge transport known as transient localization theory models[60–62]. Still, the electronic coupling between the molecules in a material directly defines the propensity of its semiconducting properties, therefore, it may be used as an indication of the CT change upon the change of the chemical composition or topology of a material. In the present study, we do not

focus on the prediction of intrinsic CT of the considered MOFs, but on the range of the electronic coupling change upon the photoconversion. Mathematically, the transfer integral, $J$, is represented as follows:

$$J_{if} = \langle \varphi_i | \hat{H} | \varphi_f \rangle, \quad (2)$$

where, $\varphi_i$ and $\varphi_f$ are the wavefunctions of the initial and final states, and $\hat{H}$ is the Hamiltonian operator representing the total energy of the system. Thus, the electronic coupling as such is a matrix element, providing a measure of the overlap between the initial and final state wavefunctions and the energy involved in the transition. It is known to be estimated using several computational techniques that can be based on different levels of theory. Among the most commonly known techniques are methods based on Koopmans' theorem, molecular orbitals (MO) and within perturbation theory[63]. Here, we have applied a molecular orbitals approach to calculate the electronic coupling between spiropyran linkers in SP-based MOFs and merocyanine linkers in MC-based MOFs. For this, MO obtained from a calculation on the isolated linker (i.e. monomer) was used as the basis set for the calculation on the dimer system, i.e. two linkers from the MOF. Then, the electronic coupling was calculated with a Löwdin orthogonalization method[64,65] using Fock- and overlap matrices of the dimer:

$$J_{ij} = \frac{\left( H_{if} - \frac{1}{2} \left( H_{ii} + H_{ff} \right) S_{if} \right)}{\left( 1 - S_{if}^2 \right)} \quad (3)$$

with

$$H_{if} = \left\langle \varphi_i | \hat{H}_{KS} | \varphi_f \right\rangle \quad (4)$$

and

$$S_{if} = \left\langle \varphi_i | \varphi_f \right\rangle. \tag{5}$$

For hole and electron transport, $\varphi_i$ and $\varphi_f$ are the HOMO and the LUMO molecular orbitals of the respective isolated monomers (linkers) and $H_{if}$ is the charge transfer integral. $\hat{H}_{KS}$ is the Kohn-Sham operator of the neutral dimer of two molecules, whereas $H_{ii}$ and $H_{jj}$ are the site energies of the two monomers. $S_{if}$ is the spatial overlap.

Considering the charge carrier mobility under hopping transport, which is often used to explain the semiconduction properties of MOFs due to rather large intermolecular distances[28,66–68], it is defined as:

$$\mu = \frac{eL^2}{k_B T} k_{CT}, \tag{6}$$

where $L$ is the center-of-mass distance between the linker molecules and $k_{CT}$ is the CT rate defined, as an example, in Eq. (1). Therefore, the CT rate and the charge carrier mobility, and thus the conductivity, depend quadratically on the electronic coupling, disregarding the reorganization energy, and deviations in the site energies. As such, the change in the electronic coupling can be used as an indication of the change in the conductivity of the system. Note that the final value of the charge carrier mobility depends on dynamic and energetic disorder in the system. The dynamic disorder is defined by vibrational flexibility of the material, leading to the change of the electronic coupling between molecules and its semiconducting characteristics. Impact of such motions, e.g. rotations, pedal motions or dipole orientational disorder, involving changes in molecular conformations of assembled molecules, was reported for various classes of materials[69–71]. Energetic (static) disorder is related to the spread or variation in the time-averaged energy levels of molecules in the material[72,73], which stems from the time-independent lack of perfect order. In our study, dynamic disorder of ordered MOFs refers to the local motions of linkers within the MOF structure. To account for it in the on-off conduction photoswitching, we calculated the electronic couplings between molecules extracted from the MD simulation at 298 K. Such an approach can directly demonstrate the deviation of the electronic coupling as a function of the vibrational flexibility of the system. For the calculation of the electronic coupling in our study, we have used DFT calculations using the Quantum Patch code as described in the "Methods" section.

Initially, we estimated the influence of the separation distance in the MOF on the electronic coupling between two linkers. In principle, this can be realized by the length of the pillar linker (see Fig. 2). Therefore, the electronic coupling between the layer linkers from adjacent MOF layers (i.e. separated by pillar linkers, see Fig. 3a, b) was initially calculated using DFT-optimized MOF structures at 0 K.

Dimers of layer linkers (as depicted in Fig. 3a, b) were extracted from the periodically optimized $Cu_2(SP)_2(dabco)$ and $Cu_2(MC)_2(dabco)$ MOF supercells using Achmol code[17]. Metal nodes were omitted and the structures were neutralized by hydrogen atoms instead. Molecules in the dimer were moved on the specific distance, defined by center-of mass distance (COM), to mimic the length of the pillar linker (marked with green arrow in Fig. 3). Electronic coupling matrix elements between linkers in such dimers, without any further geometry optimization, were calculated. In Fig. 3c, d, the distance dependence of the electronic coupling between SP- and MC-based linkers at 0 K is depicted. Similar calculations were performed also on a snapshot extracted from MD simulations (see Fig. S5), which explicitly contains the vibrational relaxation at room temperature. No significant

differences in the absolute value of the electronic coupling between HOMO orbitals of both linkers are observed (see Fig. 3c). The electronic coupling between the linkers separated with the 9 Å distance (approximately the COM, when the dabco linker is used as a pillar, i.e. 9.2 Å) equals to 8.3 meV and 9.7 meV for SP and MC, respectively. This means that the on-off conduction photoswitching is rather low and photoisomerization does not lead to a significant increase in conduction. At 10 Å separation distance we find couplings of 2.3 meV and 3.2 meV, which decrease even stronger at larger distances. Beyond a distance of 12 Å, the couplings are negligible small. This goes along with the typical distance-dependence of the electronic coupling in materials[74]. The highest coupling values of MC-based linkers correspond to intersheet distances lower than 9 Å (at 8 Å the coupling is 19.4 meV, see blue dot in Fig. 3c), however, the respective SP-based MOF could not be generated by moving the DFT-optimized dimers extracted from the periodic MOF, due to the steric obstacles and the overlaps of atoms, therefore, the on-off photoswitching cannot be established. As a result, the electronic coupling of the linkers at small distances was calculated using the structures extracted from MD simulations (see Fig. S5a) at 6.39 ns and 7.61 ns for SP and MC, respectively, where the linkers show the strongest coupling upon the structural relaxion at room temperature. In this case, the vibrational flexibility of the linkers permitted the change of the chromene unit orientation and a lower screening distance. In Fig. S5a, electronic coupling of unmodified MC is 4.6 times and 2.8 times higher than of SP, if the COM is 8 Å and 9 Å, respectively. Therefore, the on-off switching is stronger, i.e. 8-21 times higher upon photoisomerization. However, it is still rather low. The change in the on-off switching observed in MD-simulation-extracted snapshot (in comparison to the MOF at 0 K) indicates that structural motions of spiropyran- and merocyanine-based linkers impact the transfer integral between the neighboring molecules. This effect is stronger for merocyanine. It is directly related to the structurally more flexible merocyanine-based side chain, especially near the formed conjugated double bonds upon the C-O bond breakage in spiropyran (see Fig. 1).

More pronounced on-off switching is observed for electron transport, i.e., for the electronic coupling between the LUMO orbitals (Fig. 3d). Here, the electronic coupling of SP of 2 meV increases to 14.2 meV for MC, resulting in ≈7 times higher coupling values. This can result in ≈50 times higher CT rates in the case of Marcus hopping transport. Moreover, the on-off ratio is still high at the distance of 11 Å, which is not characteristic for p-conduction. Slightly lower absolute values of couplings and ratios were obtained from the distance screening calculations using the same MD-simulation-extracted MOF structures (Fig. S5b). It may be caused by the vibrational flexibility of the linkers in the material and significant changes of the electronic coupling between the LUMO orbitals as a function of molecular movements at room temperature.

Since it was previously demonstrated that vibrational flexibility of linkers in MOFs plays an important role in CT transport[17,26], we conducted more detailed analyses of MD simulations. Specifically, simulations of eight MOFs, which are $Cu_2(SP)_2(dabco)$, $Cu_2(SP-OH)_2(dabco)$, $Cu_2(SP-CHO)_2(dabco)$, $Cu_2(SP)_2(bipy)$, $Cu_2(MC)_2(dabco)$, $Cu_2(MC-OH)_2(dabco)$, $Cu_2(MC-CHO)_2$ (dabco) and $Cu_2(MC)_2(bipy)$, are analyzed below. All calculations were performed using the same computational setup utilizing workflow depicted in Fig. S1 with the flowchart explained in Fig. S2. The dimer (and for the MC case also a trimer) of the layer linkers was extracted from the central part of the $4 \times 4 \times 4$ supercell of the MOF (see Fig. 4). Visualization of the extracted pairs of molecules is depicted on the selected snapshots from the 10 ns MD-simulation runs in Fig. S6–S13. To

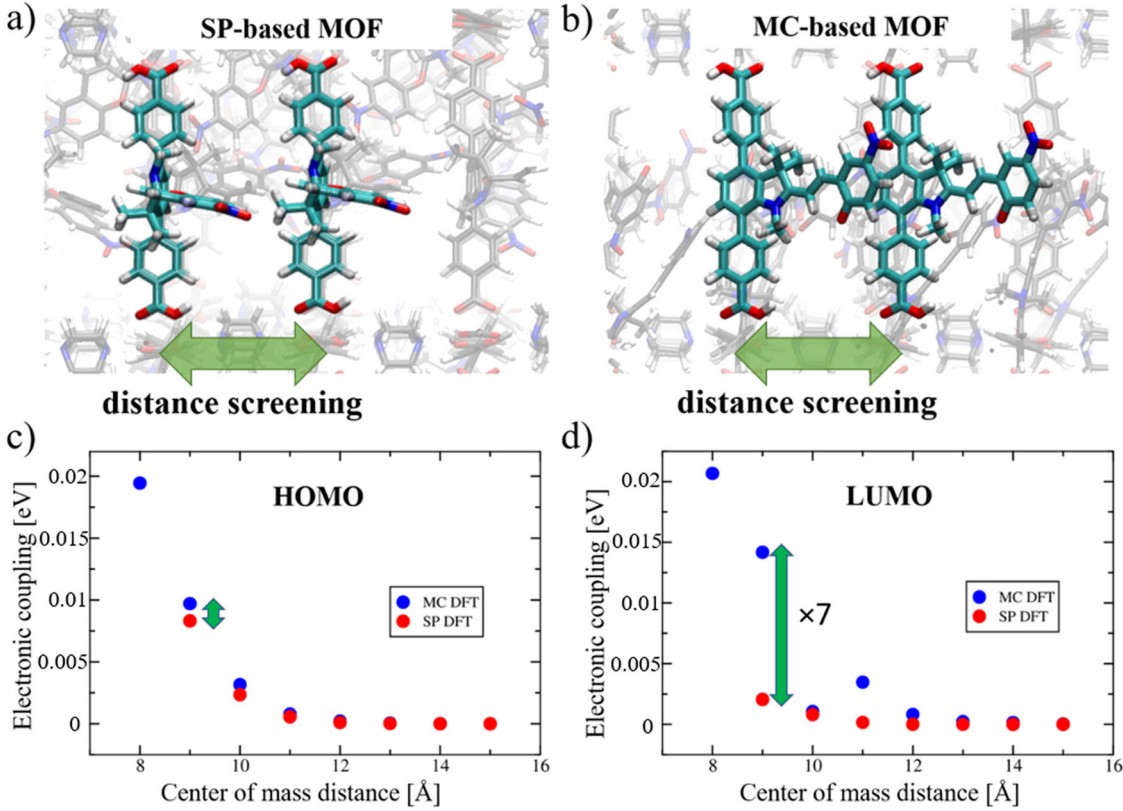

**Fig. 3 Dependence of the electronic coupling on the distance between the parallel layer linkers in the pillared-layer MOF.** Distance screening from the initially DFT optimized **a** $Cu_2(SP)_2(dabco)$ and **b** $Cu_2(MC)_2(dabco)$. The electronic coupling between **c** HOMO and **d** LUMO orbitals of two neighboring SP (in red) and MC (in blue) layer linkers extracted from the MOF and moved on a specific distance (center-of mass distance, COM) using Achmol code.

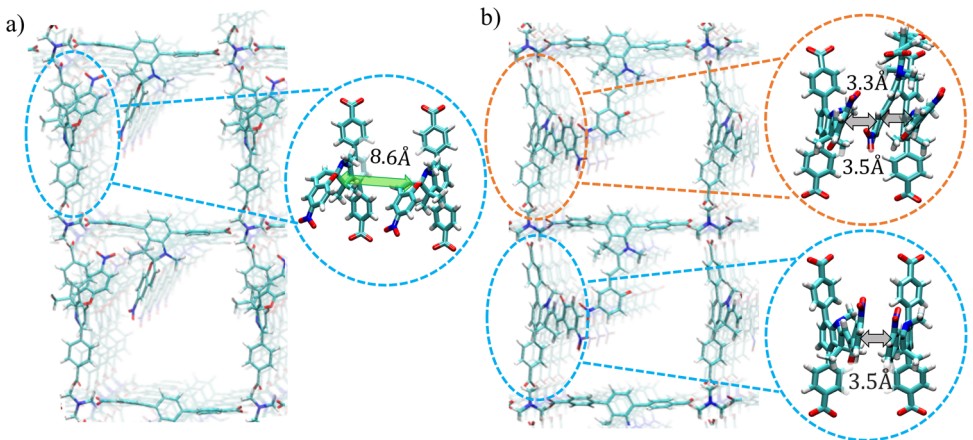

**Fig. 4 Dynamics of linkers in the MOF.** Vibrational flexibility of the layer linkers in (**a**) $Cu_2(SP)_2(dabco)$ and (**b**) $Cu_2(MC)_2(dabco)$. Due to its rigidity, the SP linkers interact only on a relatively large distances, of e.g. 8.6 Å, while the MC linkers form various π···π interactions pairs and channels. The parallel neighboring linkers, separated by the pillar linker, as depicted in the dashed circles, were extracted from the MOF MD trajectories and used for the calculation of the electronic coupling matrix elements depicted in Fig. 5. Linkers extracted for direct electronic coupling and electronic coupling via the superexchange-like mechanism are marked inside the blue- and orange-colored circles, respectively.

demonstrate the time evolution of the electronic coupling between neighboring layer linkers, couplings between extracted linkers (similarly as depicted in Fig. 3a-b, without any additional distance screening) in 250 snapshots from the last 5 ns of the MD simulation were calculated using the MO approach (Eqs. 2–5) and DFT level of theory. In the following, we analyze the on-off photoswitching between $Cu_2(SP)_2(dabco)$ and $Cu_2(MC)_2(dabco)$, see Fig. 4. Time evolution of the electronic coupling in other MOFs is depicted in Figs. S15–S17.

In comparison, to the electronic coupling between HOMO orbitals of MC linkers at 0 K (Fig. 3c), the absolute values of the electronic coupling matrix elements during MD simulation at 298 K are approximately ten times higher (i.e. even 100 meV instead of 9.7 meV, see upper panel in Fig. 5a). This happens due to rather intense motions of the MC-based side chain of the layer linker. It allows to form attractive interaction patterns (see snapshot "3" and "4" in Fig. S7), permitting a better overlap of molecular orbitals, thus increasing the transfer integral. Owing to the fact that the

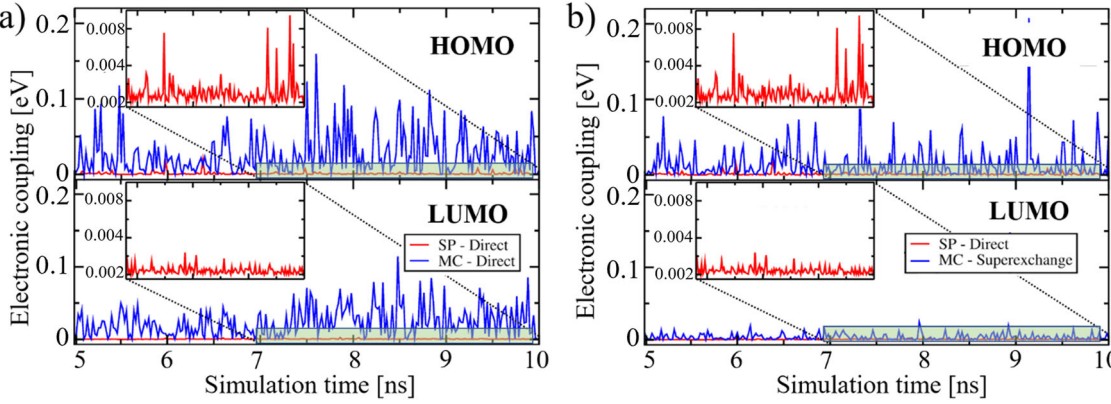

**Fig. 5 Electronic coupling between linkers.** The time evolution of the electronic coupling between HOMO and LUMO orbitals of SP (red) and MC (blue) linkers at 298 K, see labels. **a** Direct coupling calculated between neighboring linkers extracted from Cu₂(SP)₂(dabco) and Cu₂(MC)₂(dabco), see extracted pair of molecules in blue circles in Fig. 4. **b** Comparison of direct coupling between SP linkers and the total electronic coupling with superexchange-like process (defined in Eq. 7) between MC linkers (shown in orange circles in Fig. 4).

HOMO of MC is delocalized over the entire molecule with the highest electron density on the substituted phenyl ring (see Fig. 1), such interaction patterns lead to a significant temporary increase of the coupling values up to 100 meV. The electronic coupling between SP-based linkers is slightly reduced in comparison to the values obtained for DFT-optimized MOF structures: couplings in MD snapshots at 300 K are mostly around 1.15 meV, see inset in Fig. 5a. Temporary increases of up to 8 meV are also noticeable. Similar values were obtained at 0 K. Such an observation is based on the fact that the SP side chain is rather rigid and compact, hindering a reorientation to establish a significant overlap of molecular orbitals. At the same time, the orbital localization differs from the merocyanine case, therefore, structural motions do not impact significantly the electronic coupling between SP since indoline and the chromene moiety are rather less labile. The difference in the vibrational flexibility of the isomers and the difference in their electronic properties permits an increase of the electronic coupling between the HOMO orbitals of 23 times during the on-off SP⇌MC-switching, considering the average values over 5 ns. Here, the arithmetic mean values of the coupling for MC and SP is 26.79 ± 27.77 meV and 1.15 ± 2.03 meV (see Table S1), while the geometric mean is 15.27 meV and 0.55 meV with the geometric standard deviation of 3.75 and 3.56, respectively. Due to the significant fluctuations of the electronic coupling over time at 300 K, there is no straightforward way to calculate the effective CT rate (Eq. 1) using the average value of the coupling. Therefore, the approximated CT rate may possibly increase by 530 or even higher (see the multitude of couplings for MC over 50 meV in the upper panel in Fig. 5a), when the separation distance between the linkers is around 9 Å (corresponding to a dabco pillar).

A similar increase of the electronic coupling is characteristic for electron transport (bottom plot in Fig. 5a). However, the absolute value of coupling between LUMO orbitals of MC is slightly lower than of HOMO orbitals (around 50 meV, the average value is 23.25 ± 20.26 meV, as listed Table S1). We have to point out that in both cases, i.e. for HOMO and LUMO orbitals of MC, the electronic coupling is strongly varying, which would impact the overall CT in a MOF, increasing possible charge localization. At the same time, the temporary high MO overlaps permit possible charge transfer, however, the transport would be of highly anisotropic and dynamic nature.

The electronic coupling between LUMO orbitals of SP-based linkers is also lower and shows a large decrease in the average coupling (see inset in Fig. 5a): 0.36 ± 0.39 meV instead of 1.15 ± 2.03 meV. Therefore, the approximated on-off ratio of the CT

rates (considering the average values over 5 ns) in n-conduction may be even 65² ≈4200 (see Table S1). This is caused by large differences in the orbital overlap of the neighboring SP-based and MC-based linkers, which are directly connected to the dynamics of the linkers in the MOF explained above. Moreover, the increased molecular flexibility of MC permits the formation of stable π⋯π interactions and π-stacking channels with intermolecular distances of ca. 3.3–3.5 Å between the linkers (Fig. 4b). Such interactions additionally enhance the orbital overlap (i.e. the electronic coupling). The interatomic distances between two SP linkers are often higher than 8 Å (Fig. 4a), while the distances between the MC linkers in the MOF with the same pillar linker (i.e. for the same distance between the MOF lattice planes; here dabco was used with a lattice distance of 9.2 Å) are much lower. We have to point out that due to its very high flexibility, the MC-isomer can also form π⋯π channels, as visualized in Fig. 4b (marked in gray). In this case, not only layer linkers directly separated by the pillar interact, but also linkers located on other axes. This demonstrates that not only direct electronic coupling between MC is relevant, but also a coupling modulated by a superexchange-like mechanism[28]. In this mechanism, the electronic coupling between two linkers is enhanced by the third linker, playing a role of a mediator. In this sense, the coupling between layer linkers, located on the parallel planes (i.e. in a dimer that we consider in this work) is larger when the linker from another axis which mediates the transport. For that, three coupling values are calculated, i.e. between all possible dimer structures forming a trimer, therefore, the total electronic coupling, including the superexchange-like coupling, is obtained using Eq. (7)[28]:

$$J_{gg'}^{tot} = J_{gg'}^{dir} + J_{gg'}^{sx} = J_{gg'}^{dir} + \sum_h \frac{J_{gh}^{dir} \times J_{hg'}^{dir}}{\Delta E_{sx} + \frac{1}{2}\lambda_h}, \tag{7}$$

where $\lambda$ is the reorganization energy, $J_{gg'}^{dir}$ is the direct electronic coupling from site $g$ to $g'$, $J_{gg'}^{sx}$ is a coupling matrix element of a superexchange-like process (i.e. charge hop from $g$ to $g'$ via $h$), $h$ is the virtually occupied (intermediate, mediating) host state.

The coupling originated from the superexchange-like process ($J_{gg'}^{sx}$) would then depend on the direct coupling between $g$ to $h$ and $h$ to $g'$ and reorganization energy. The denominator $\Delta E_{sx}$ defined as:

$$\Delta E_{sx} = E_h - \frac{1}{2}\left(E_g + E_{g'}\right), \tag{8}$$

indicates the difference in the site energies of the virtual (host) state. It is skipped in the present study since the mediator molecule is of the same type as the molecules involved in the charge hop.

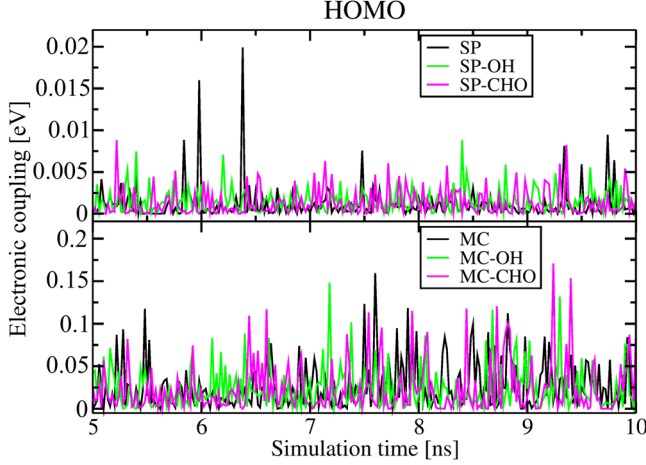

**Fig. 6 The change in electronic coupling upon linker functionalization.**
The time evolution of the electronic coupling between HOMO orbitals of unmodified SP and MC linkers in $Cu_2(SP)_2(dabco)$ and $Cu_2(MC)_2(dabco)$ (in black), and their OH- and CHO-modified analogs from $Cu_2(SP-OH)_2(dabco)$ and $Cu_2(MC-OH)_2(dabco)$ (in green) and $Cu_2(SP-CHO)_2(dabco)$ and $Cu_2(MC-CHO)_2(dabco)$ (in violet), respectively. Only direct electronic couplings are considered.

From Fig. 5b, we see that the electronic coupling between LUMO orbitals of MC, which includes the impact of the superexchange-like process (through space), in $Cu_2(MC)_2(dabco)$ is lower than direct hopping between MC units in the MOF (depicted in blue in the lower panel in Fig. 5a). However, the coupling via the superexchange-like mechanism (see Eq. 7) between HOMO orbitals is still relevant: the average coupling is 14.5 meV with temporary couplings of 50-70 meV that may additionally improve the on-off photoswitching. Frequently observed high coupling values in Fig. 5b indicate that the mediating linker (from another axis than the consecutive linkers considered in the direct coupling, see structures in Fig. 4b) fills $\pi\cdots\pi$ interactions with linkers and takes the position in between them to maximize the attractive interaction. The possibility for charge hops in this case increases, improving the conduction processes in the MC-based MOFs. Such an effect was not observed for SP-based linkers due to their higher linker rigidity.

To estimate the change in on-off SP⇌MC switching upon linker functionalization with OH and CHO groups, similar MD simulations and electronic coupling calculations were performed for $Cu_2(SP-OH)_2(dabco)$, $Cu_2(MC-OH)_2(dabco)$ and $Cu_2(SP-CHO)_2(dabco)$, $Cu_2(MC-CHO)_2(dabco)$, respectively. The dynamical evolution of the electronic coupling matrix elements after the chemical modification of the linkers is depicted in Figs. 6 and S14. The comparison of the electronic coupling upon SP⇌MC photoconversion is demonstrated in Figs. S15, S16. In general, linker functionalization with OH leads to a slight increase of the orbital overlap of HOMO and LUMO of SP-based linkers (see data marked in green and black in the upper panel in Figs. 6 and S14). The average value from 250 snapshots is 1.5 meV and 1.0 meV, respectively, while 1.15 meV and 0.36 meV for unmodified linkers. At the same time, the trend in electronic coupling between OH-functionalized MC-based linkers decreases: 23.6 meV and 14.2 meV instead of 26.8 meV and 23.2 meV for HOMO and LUMO, respectively. It is also visible in the lower panel in Figs. 6 and S14 (i.e. couplings in green are slightly smaller than couplings in black). This results from less efficient $\pi\cdots\pi$ interactions and $\pi$-stacking channels due to the OH-functionalization (see Fig. S9). Thus, the on-off photoswitching of n-conduction in OH-functionalized MOFs is lower, while the

p-conduction should be less affected. Still, the electronic coupling of HOMO orbitals of MC-based linkers in $Cu_2(MC-OH)_2(dabco)$ is relatively high (note the one order of magnitude difference in the scale of the electronic coupling in Fig. 6). This observation demonstrates a significant (promising) level of conduction switching. Still, coupling values fluctuate intensively that may impact the measured conductivity of the material[60].

Similar effects are observed for the linkers modified with the aldehyde groups (data in violet in Fig. S6): the average electronic coupling (from 250 snapshots) between HOMO orbitals of MC-based linker is 24.1 meV with less frequent increases of the coupling values (see Fig. S16 for clarity). At the same time, the coupling of the LUMO orbitals is better than in $Cu_2(MC-OH)_2(dabco)$, i.e. 21.3 meV (see Figs. S14 and S16), but still slightly lower than for the unmodified linkers. Frequent couplings up to 100 meV, as shown for $Cu_2(MC)_2(dabco)$, are also possible, which is connected to the similar $\pi\cdots\pi$ interactions between neighboring linkers (see Fig. S11). Considering that coupling between orbitals of SP linkers is higher in both $Cu_2(SP-OH)_2(dabco)$ and $Cu_2(SP-CHO)_2(dabco)$ than in $Cu_2(SP)_2(dabco)$, the final on-off ratio is ~20% lower.

As suggested by the distance screening of the electronic coupling, shown in Fig. 3, the most efficient on-off switching can be realized by the pillar length of 8-11 Å (like for dabco-pillar linkers). Due to significantly lower values of the electronic coupling at larger distances, both conduction and on-off switching is irrelevant. To verify this fact also at 298 K, MD calculations with bipy pillar (the respective COM distance of 13.7 Å) were performed. Due to larger accessible free space around the neighboring linkers in such a MOF (see Figs. S12, S13), flexible molecules as MC, linked to the core of the ditopic linker, are freely and randomly rotating within the unit cell, restricting charge hopping within the material. However, after some time (around 7 ns of MD simulation) the open fragment of MC twists with increased intermolecular coupling (see upper plot in Fig. S17). Even if some on-off switching can be observed, the high flexibility of the MC system results in low average electronic coupling (while the average electronic coupling of SP-based linkers is relatively high: 2.7 meV and 13.3 meV between HOMO and LUMO orbitals, respectively). Therefore, bipy-pillared MOFs are not efficient candidates for on-off conduction photoswitching materials, based on spiropyrans.

As pointed above, the on-off conduction switching significantly depends on the electronic coupling, but for spiropyran MOFs we have observed that it may be also connected to the differences between the energy of the frontier orbitals of SP and MC (see Table S2, Figs. 7 and S18). If the orbital energy difference $\Delta E_{SP-MC}$ is high, the on-off switching is, in general, improved. For example, $\Delta E_{SP-MC}^{HOMO}$ and $\Delta E_{SP-MC}^{LUMO}$ of plain layer linkers (chemical formula in Fig. 2) are −0.35 eV and +0.71 eV, while of the OH- and CHO-functionalized are −0.11 eV, +0.43 eV and −0.02 eV, +0.40 eV, respectively. MOFs with the unmodified linkers have shown on average better electronic coupling (see Fig. 6) and on-off ratios. This suggests that higher on-off conduction switching can be modulated not only via intermolecular distances, but also by the design of spiropyran linkers and their orbital energies and electron density occupation. As to our knowledge, there is no direct explanation of the change in the HOMO (or LUMO) value of two photoisomers and the change of the electronic coupling between them.

To demonstrate the impact of other electron donating and accepting functional groups on the change of the orbital parameters, several candidates have been investigated. Their HOMO and LUMO orbitals are visualized in Figs. 7 and S18, respectively. High $\Delta E_{SP-MC}^{HOMO}$ and $\Delta E_{SP-MC}^{LUMO}$ were found also for $SP-NH_2-C_2H$ and $MC-NH_2-C_2H$, i.e. +0.31 eV and +0.88 eV, respectively (see

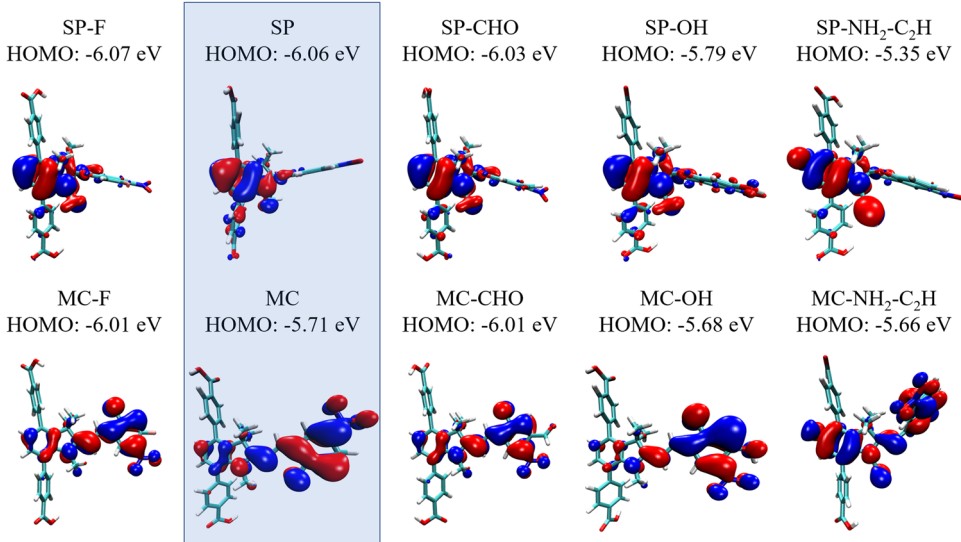

**Fig. 7 Visualization of HOMO orbitals of linkers considered in the present study (see Fig. 2 for clarity).** Orbitals are given with the respective orbital energy. Isovalue of 0.02 a.u. was used for visualization. The unmodified SP and MC linkers are highlighted in light blue. MOFs with SP, MC, SP-OH, MC-OH, SP-CHO and MC-CHO layer linkers were simulated.

Table S2). The on-off ratio of the electronic coupling as a function of lattice distance in this case is also high (see Fig. S19). However, the absolute values of coupling are approximately half as large as for the plain layer linkers, depicted in Fig. 3. Based on all analyses, we conclude that the change in the on-off conduction ratios upon spiropyran linker functionalization is modulated by several effects: 1) the electron density occupation of the HOMO and LUMO orbitals, 2) the change in the energy of the frontier orbitals upon SP-to-MC isomerization, 3) the vibrational flexibility of the linkers in the MOF and 4) the possibility of π···π-stacking.

Finally, we have to point out that there may be several factors reducing the overall on-off conduction in the spiropyran-based MOF demonstrated in this study. Among them is the degree of the photoconversion between both isomers, the localization of the photoswitchable side chains with respect to each other and the presence of defects in the fabricated MOF. Since we considered 100% photoconversion of linkers in the pillared-layer MOFs, defect-free material and the starting location of the side chains of the layer linkers in the MOF as pointing in the same direction, the on-off ratios reported should be treated as the maximal possible values that could be achieved for the photoswitching in spiropyran-based MOFs.

To experimentally reproduce the theoretical findings, the syntheses of surface-mounted MOF (SURMOF) films with the targeted structures (see Fig. 2) were performed in a layer-by-layer fashion. With this established method, i.e. by subsequently immersing the substrates in the ethanolic solutions of the copper-acetate-metal-nodes and the linker molecules, SURMOFs with various pillared-layered structure have been prepared, see e.g. refs. [75–77]. With the aim of synthesizing the MOF films with the designed structures, many parameters were varied and optimized, however, the obtained X-ray diffraction data show that the prepared films have no high crystallinity. Further experimental optimizations are essential and will be presented elsewhere.

## Conclusions
Using multiscale modeling and automated workflows, several spiropyran-functionalized MOFs were generated in silico, optimized and characterized. Combining density functional theory and molecular dynamics simulations, we were able to demonstrate the high potential of pillared-layer MOFs with spiropyran-based layer linkers for conduction photoswitching. We find the electronic coupling between the HOMO and LUMO orbitals of the linkers in such MOFs increase on average more than 23 and 65 times upon spiropyran-to-merocyanine isomerization. This indicates an increase in the charge transfer rate (which depends quadratically on the electronic coupling) by approximately more than 530 and 4200 times. The degree of conduction switching is significantly enhanced by dynamical structural changes of the switches at room temperature. These effects are very distance dependent, therefore, only pillar linkers of specific size are suitable for conductance switching. Modification of the spiropyran-moiety of the linker via additional functionalization with electron donating and withdrawing groups were demonstrated to impact the energy of the frontier orbitals and the orbital occupation (i.e. electron density delocalization), changing the on-off switching performance. The combination of these effects offers a wide range of possibilities to tune the materials towards even higher on-off ratios. Thus, spiropyran-based layer linker in dabco-pillared MOF is a potential material with promising light responsive properties.

The study provides a deeper understanding of the photoswitching in SP-functionalized MOFs. It reports potential on-off ratios upon photoisomerization, however, several further effects may impact on the final conduction switching that were not considered in the present work, such as the isomerization yield between SP and MC and the presence of defects. At the same time, this study provides guidelines which can be transferred to other switchable molecules like spirooxazine and diarylethene and to other classes of materials, in particular framework materials like covalent organic frameworks, where the mentioned factors may play a different role. In that way, it contributes to the fast development of smart materials.

## Methods
**Computational workflow.** MOFs in the present study were modeled and characterized using a previously developed automated workflow[17] in SimStack workflow client[17,78] (see Fig. S1). Several Workflow active Nodes (WaNos), programmed to perform a specific type of calculation, were coupled together to enable MOF model generation, DFT optimization (or pre-

optimization) and parameter passing from the DFT output data to all-atom MD input data to perform further simulations at room temperature. Two main dependencies of the electronic coupling were studied. First, the electronic coupling was calculated as a function of the separation distance between the layer linkers in the MOF. For that, either plane-wave-DFT-optimized MOFs or the MD-simulation-extracted snapshots were used. Second, the time evolution of the electronic coupling as a function of linker vibrational dynamics at specific intersheet distance, controlled by the length of the pillar linker (dabco or bipy), were calculated using snapshots from MD. In both cases the electronic coupling was calculated using MO approach (Eqs. 2–5). For that Quantum Patch method[79] on the DFT level (B3LYP[80,81]/def2-SVP[82]) was used. The detailed description of all steps performed and principles of the workflow is given in SI. Below, we explain the computational setup used in WaNos of the workflow and additional procedures that were handled to analyze data we obtained.

**Density functional theory calculations**. Optimization of all linkers considered (DFT-Turbomole WaNo in Fig. S1) was obtained using B3LYP functional with def2-TZVP basis set with the Grimme D3 dispersion correction[83] in TURBOMOLE[84], version 7.4.1. Linkers were used to build an initial MOF model (LCmaker and AuToGraFS WaNo), which was further optimized using Perdew-Burke-Ernzerhof (PBE[85,86]) functional with plane wave energy cutoff of 500 eV over a k-point grid with dimensions of $2 \times 2 \times 2$ in Vienna Ab initio Simulation Package (VASP[87,88]), version 5.4.4 (DFT-VASP WaNo). The electron–ion interactions were described by the PAW[87](projector augmented wave) potentials. Full geometry optimization of $Cu_2(SP)_2(dabco)$ and $Cu_2(MC)_2(dabco)$ was performed using the conjugate gradient algorithm until the electronic self-consistent-loop reached convergence of $10^{-4}$ eV and the ionic relaxation loop of 0.01 eV. The Tkatchenko-Scheffler method with iterative Hirshfeld partitioning[89–91] was used to include dispersion correction. The pre-optimization of all other MOFs, utilized as an input for MD simulations, was performed with 100 ionic steps with the break ionic relaxation criterium of –0.02 eV. Grimme D2 dispersion correction[92] was used to accelerate pre-optimization.

For the distance screening with the simultaneous electronic coupling calculations between layer linkers in the MOF two neighboring linkers were extracted from the optimized periodic MOF structure using Achmol[17,93] and moved to a sequence of intermolecular distances, incrementing every 1 Å with an in-house script. The electronic coupling was calculated for linkers separated by a distance in the range of 8–15 Å. Lower distance values were not considered due to the overlap of atoms between the molecules, whereas at higher distance values the couplings are lower than 0.01 meV. A similar approach was used for the calculation of the electronic coupling between molecules extracted from MD snapshots (Fig. S5)

**Molecular dynamics simulations**. DFT-optimized (or pre-optimized) MOF structures were also used as an input for MD simulations of MOFs using Large-scale Atomic Molecular Massively Parallel Simulator[94] (LAMMPS) at 298 K (lammps-interface and LAMMPS WaNo)[95]. MD runs were performed using UFF4MOF[96,97] force field. The $4 \times 4 \times 4$ MOF supercells were simulated with a timestep of 1 fs. The equilibration was done using an NVE ensemble and a Langevin thermostat[98] for 0.3 ns, whereas the production MD runs were performed using the canonical (NVT) ensemble and Nosé-Hoover thermostat[99] for 10 ns. The position of metal ions was re-centered and constrained to their center of mass in all directions based on the DFT data. All

other parameters were taken as specified in lammps-interface[95], e.g. real unit with LJ and Coulomb interactions cut-off 12.5 Å, as well as the use of thermostats for MOFs. After the detailed analysis of MD trajectories, the pair of layer linkers, representing the overall dynamics of the MOF components, was selected and 250 snapshots of interacting linkers (i.e. neighboring linkers as schematically shown in Fig. 3) (every 20 ps) from the last 5 ns MD simulations were extracted. Molecules were automatically hydrogenated ($COO^-$ groups that were linked to Cu metal nodes in a MOF) using in-house scripts using Atomic Simulation Environment[100] and openbabel[101], and the electronic coupling between the molecules were calculated as an overlap of molecular orbitals explained previously. The summary of all calculations performed is depicted in Fig. S2.

## Data availability

The datasets generated during and/or analyzed during the current study are available from the corresponding author upon reasonable request. The datasets are also available via the NOMAD repository under https://doi.org/10.17172/NOMAD/2023.12.04-3. Supplementary information, containing the description of the automated workflow, the visualization of selected snapshots from MD and molecular orbitals, and the time evolution of the electronic coupling in MOFs from MD (not presented in the main body) are available as a separate document.

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

## Acknowledgements
This research was funded by the Deutsche Forschungsgemeinschaft (DFG) via SPP 1928 COORNETs (HE7036/6 and WE1863/37) and via GRK 2450 "Scale bridging methods of computational nanoscience". H.P.H. acknowledges funding by the Deutscher Akademischer Austauschdienst (DAAD). Authors acknowledge the Virtual Materials Design initiative (VirtMat) funded by KIT. This work was performed on the HoreKa supercomputer funded by the Ministry of Science, Research and the Arts Baden-Württemberg and by the Federal Ministry of Education and Research and bwHPC JUSTUS 2 funded by the state of Baden-Württemberg (Germany). Authors want to thank Ali Deniz Özdemir for fruitful discussions. Funded by the European Union (ERC, DYONCON, 101043676). Views and opinions expressed are, however those of the author(s) only and do not necessarily reflect those of the European Union or the European Research Council. Neither the European Union nor the granting authority can be held responsible for them. We acknowledge support by the KIT-Publication Fund of the Karlsruhe Institute of Technology.

## Author contributions
Conceptualization: W.W., L.H., and M.K.; methodology: M.M., W.W. and M.K..; software: M.M.; validation: M.M., H.P.H., Y.J. and M.K.; formal analysis: M.K.; investigation, M.M., H.P.H., Y.J. and M.K.; resources, W.W. and L.H.; data curation, M.M. and H.P.H; writing – original draft preparation: L.H. and M.K; writing – review and editing: M.M., H.P.H., Y.J, W.W., L.H., and M.K.; visualization, M.M. and H.P.H.; supervision: W.W., L.H. and M.K.; project administration: L.H. and M.K.; funding acquisition: W.W. and L.H. All authors have read and agreed to the published version of the manuscript.

## Funding

## Competing interests
The authors declare no competing interest.
