## [Peer Review File · Communications Chemistry]

Reviewers' comments:

Reviewer #1 (Remarks to the Author):

The work of Mostaghimi and coworkers presents the computational design and analysis of various MOF with photoswitchable conductivity using the spiropyran photoswitchable ligand. Molecular design based on the interchromophore distance and using different substituent groups allows to tune the photoswitchable properties of the MOF to achieve several orders of magnitude of difference in the charge carrier mobility between the two isomers. Throughout computational analysis based on DFT structure optimization, FF molecular dynamics and electronic coupling calculations is done for 8 MOF structures based on Cu-paddlewheel units connected through the spiropyran photoswitch forming layers that are pillared connected with ligands of different sizes: dabco and bipyridine. The conclusions of the work are well supported by the simulations which are based on robust theory of charge transport in molecular materials and thermal effects introduced by real time MD. This work is certainly interesting for the materials chemistry community working in the understanding and molecular design of functional materials. There are however some points that need clarification in the manuscript.

-In the abstract and introduction, it is mentioned that the trial and error approach is highly inconvenient in the design of MOF and photoswitchable materials. In this context it is not clear how do the authors choose the specific setup of copper paddlewheel metal nodes connected in layers by the photoswitch and pillared by an auxiliary ligand. Is this setup is based on previous work? Or there was some rationale for this setup?

-For the results shown in Figure 3 and the discussion in page 6, how the different distances between the SP and MC linkers is achieved in the DFT optimizations of $\text{Cu}_2(\text{SP})_2(\text{dabco})$ and $\text{Cu}_2(\text{MC})_2(\text{dabco})$ is not explained. In page 8 it is specified that the dabco ligand gives rise to 9.2 Å separation, and in page 9 the bipyridine ligand results in 13.7 Å separation. However, how the other distances are obtained is not clear.

-In page 6, it is mentioned that “electronic coupling of linkers on the smaller distance was calculated using structures extracted from MD simulations”. Then at page 7 it is mentioned “Slightly better on-off photoswitching for linkers extracted from MD is found to be after OH-modification”, and at the end of the paragraph: “Electronic couplings mentioned were calculated only for one snapshot from MD, thus more sophisticated analysis is necessary.” Why only one snapshot for the OH case? Which snapshot was chosen and why? Are these MD at 300 K? How long are these dynamics?

-second paragraph in page 6 starts: “Since it was previously demonstrated that vibrational flexibility of linkers in a MOF may play an important role in the CT transport in a material, 10 ns MD simulations of $\text{Cu}_2(\text{SP})_2(\text{dabco})$, $\text{Cu}_2(\text{SP-OH})_2(\text{dabco})$, $\text{Cu}_2(\text{SP-CHO})_2(\text{dabco})$, $\text{Cu}_2(\text{SP})_2(\text{bipy})$, $\text{Cu}_2(\text{MC})_2(\text{dabco})$, $\text{Cu}_2(\text{MC-OH})_2(\text{dabco})$, $\text{Cu}_2(\text{MC-CHO})_2(\text{dabco})$ and $\text{Cu}_2(\text{MC})_2(\text{bipy})$ were performed for a simulated temperature of 300 K...” Do these include the previously discussed dynamics of $\text{Cu}_2(\text{SP})_2(\text{dabco})$, $\text{Cu}_2(\text{SP-OH})_2(\text{dabco})$, $\text{Cu}_2(\text{MC})_2(\text{dabco})$ and $\text{Cu}_2(\text{MC-OH})_2(\text{dabco})$? Then later in the same paragraph: “In the following we analyze on-off photoswitching between $\text{Cu}_2(\text{SP})_2(\text{dabco})$ and $\text{Cu}_2(\text{MC})_2(\text{dabco})$ ” Are these the same dynamics discussed before? Or different ones?

-starting page 9: "To estimate the change in on-off $SP \rightleftharpoons MC$ switching upon linker functionalization with OH and CHO groups, similar calculations were performed for $Cu_2(SP-OH)_2(dabco)$, $Cu_2(MC-OH)_2(dabco)$ and $Cu_2(SP-CHO)_2(dabco)$, $Cu_2(MC-CHO)_2(dabco)$, respectively. The dynamical evolution of the electronic coupling matrix elements is depicted in Figure S11-S12" Here again is not clear if these are different dynamics or it is referring to the ones previously mentioned in page 6?

-I am not sure if the last paragraph mentioning the experimental attempt to crystallize and characterize the proposed systems deserves 9 lines since, as the authors mention themselves, "Further experimental optimizations are essential and will be presented elsewhere." Moreover, experimental method details are not given. The authors may consider to remove this part and shortly mention in the conclusions that experimental validation is ongoing.

Reviewer #2 (Remarks to the Author):

In this manuscript Heinke, Kozłowska and coworkers report on a modelled spiropyran-based Metal-Organic Framework. This are very nice results and a good fundamental study, which surely deserves publication in Commun. Chem.

There are however, some points that should be improved prior to acceptance:

1) Figure 1: Please increase the resolution. Moreover, Figure 1 closely resembles Figure 4 of 10.1002/anie.201811458. The authors should be more careful about this!

2) I could follow the theoretical part of the manuscript and during that I was really looking forward to the experimental results. At the end of the manuscript, only a paragraph was reported on the experimental findings. Furthermore, it turned out that the theoretical prediction could not be reproduced, due to the lack of crystallinity...

Finally, where are the experimental results? I was not able to find it, please include them. It seems to me that both groups were disappointed and simply did not add it. Why?

3) Some Typos:

- a. 79 the quote is before the point
- b. 145 the quote is before the point
- c. 186 were calculated

Reviewer #3 (Remarks to the Author):

In the current manuscript, Mostaghimi et al. describe multi-scale simulations used to analyze the conduction switching in spiropyran-based MOFs. The topic is interesting and timely, but in my opinion the manuscript is not publishable in its current form. Besides some potentially rather serious general aspects and a rather superficial description of the methodology, my primary concern is that the model used to estimate electronic conductivities might very well not be suitable for the materials at hand due

to their strongly varying degree of dynamic disorder.

Major general aspects:

- In the simulations the authors apparently assume that upon photoexcitation 100% of the chromophores switch between the spiropyran and the merocyanine form. At least this would be my guess, as (like many other aspects) this seems to be never explicitly described in the manuscript. The authors ought to comment on whether this is a realistic assumption. In this context they also should consider that in the case of mostly one-dimensional charge transport along chromophore stacks (again I guess that this is what they assume here) a single “defect” (=non-switched molecule) can have a tremendous impact on the material’s charge-transport properties as the charge has hardly any chance to bypass that effect.

- I wonder, whether the assumed structures are realistic – in particular, considering the rather long side-chains of the linkers, can the authors be sure that the conformation they studied is the only conceivable one? In view of the employed periodic boundary conditions it seems that they assumed that the side chains in every parallel linker points in roughly the same direction (an aspect that I again never found to be discussed explicitly). Can the authors exclude that the side-chains of consecutive linkers point in very different (maybe even opposite) directions. I suppose that this would completely change and presumably diminish charge transport, but would possibly not show up in MD simulations running only for a few (ten) nanoseconds when the initial configurations consists of essentially parallel side chains (i.e., there might not be enough time for the side-chains to flip)

- The introduction is rather lengthy, more in the style of a review and not exactly to the point. In contrast, the “Results and Discussion” section is rather hard to follow. While I am not so much concerned about the lengthy introduction, a more accessible “Results and Discussion” section would be highly desirable.

- In the “Results and Discussion” section, a clearer explanation for why certain effects are observed would be desirable (for example, for the 10-fold increase of the coupling matrix mentioned in line 265).

- In lines 343 ff the authors try to extract a connection between the energies of the frontier orbitals and the electronic coupling, while not providing a sufficiently clear explanation of the origin of that connection. Thus, I wonder, whether this connection is a pure coincidence, whether there are just correlations or, whether there is really a causality in the connection between frontier orbital energies and electronic couplings?

Major technical aspect:

The analysis of charge transport is based on Marcus-type hopping rates. Considering the tiny electronic couplings between the different chromophores, I suppose assuming pure hopping transport is fair enough. What, however, could be highly problematic is that for calculating the charge transfer rates, the authors appear to use the average electronic couplings extracted from dimers found in the MD trajectories. To my understanding, just taking the average value and completely disregarding the degree

of variation of the transfer rates is inappropriate and will produce even qualitatively incorrect trends for systems like the present ones in which not only the average value of the coupling, but also its degree of variation appears to change significantly from system to system. What is actually quite confusing in this context is that in the method section the authors cite Refs. 65, 66, and 67 as papers on which their calculation of electronic couplings is based. In these papers the carrier mobility is explicitly correlated with the energetic disorder, which according to equations (1) and (2) is not done here!

In this context, I wonder, whether adopting the simple model to account for energetic disorder used in the above references would actually suffice for the highly anisotropic transport present in the studied systems. To consider also anisotropies, the authors might want to take a closer look at the dynamic disorder model developed over the years by Fratini et al.. In fact, some years ago Troisi, Fratini et al. published a paper in Nature Materials in which they not only alluded to the crucial (negative) implications of highly anisotropic transport, but in which at some point they also claimed that the magnitude of the transfer integral becomes irrelevant for charge transport in organic semiconductors in a dynamic disorder scenario. If one takes a careful look at the Supporting Information of that paper, one realizes that this statement is apparently true, when the absolute magnitude of the variation of the electronic coupling is proportional to the value of the electronic coupling. What this shows is that the degree to which the transfer integral varies over time is as important for the transport properties as the average value of the electronic coupling. Therefore, in cases in which not only the electronic coupling changes between different systems but also its variation over time it is simply not sufficient to only consider the average value.

To cut a long story short, I have serious concerns that analyzing trends in charge transport in the systems considered here using equations (1) and (2) is strongly misleading and more sophisticated models (possibly even going between what has been described in Refs. 65-67) need to be employed.

Not sufficiently detailed technical description:

- For calculating Figure 3, what is the actual structure of the model system that the authors use to increase the center of mass distance between the chromophores? Do they only consider the linkers, disregarding the metal nodes and the “pillars”? Do they then only consider linker dimers and if so do they leave their structures like in the bulk system at the equilibrium distance or do they optimize the atomic positions for every distance? In fact, I would exclude that in the actual systems the structures stayed the same, if one increased the distance.

- No details are provided on how dimer structures were extracted from molecular dynamics runs. Did the authors only extract dimers stacked on top of each other and under which circumstances were trimer structures (like those shown in Figure 4b) extracted?

- The use of the term “superexchange” is somewhat misleading. This term is borrowed from exchange coupling in ferromagnets, when the spins of magnetic metal ions are coupled e.g. via non-magnetic oxygen ions. Thus, even if the term superexchange has been used before in the context of charge transport, it is kind of a misnomer, but what is more relevant is, what the authors actually mean by it: For a structure like the one shown at the top panel of Figure 4b, do they mean that a charge carrier does two independent hops between neighboring Pi-systems (where I would not see any relation to

superexchange) or does it directly hop from the leftmost to the rightmost Pi-system with the central ring only acting as “mediator” (which one might consider as a Coulomb-coupling equivalent to superexchange).

- In the methods section, the authors mention the SimStack workflow (also depicted in Figure S1). How does workflow correlate with the description of the methodology in the lines after line 406. Is the procedure described there performed after the SimStack simulations, is it part of it. A much more clear and consistent description would be highly desirable.

- In fact, the entire modelling procedure in the methods section is described not very clearly. For the sake of reproducibility, a more coherent description of what has actually been done under which circumstances and to obtain which quantities would be needed.

- There are many ways to calculate the electronic coupling between frontier orbitals and there is a bunch of literature on pros and cons of the different approaches. Considering the high importance of the electronic couplings for the present manuscript, the authors should provide a consistent description of which of them they actually chose (merely citing an approach developed to calculate charge carrier mobilities in general is not sufficient for understanding of how, in detail, the electronic couplings have been calculated).

- Why did the authors switch between functionals and between van der Waals corrections in the different steps of their simulations. This appears inconsistent and can be done only, if a proper justification is provided.

- How large were the supercells assumed in the MD runs and did the authors observe any impact of the supercell-size on their results? How did they heat their structures? What thermostats were used? etc

Regarding the data availability, I understand that journals do not yet insist on all calculations being uploaded to dedicated databases. Still with such databases readily available, I would consider it as “good scientific practice” to make simulation data of scientific papers publicly available.

Dear referees,

We thank you for your time and effort in reviewing our manuscript COMMSCHEM-23-0153. We are thankful for the appreciation of our work and for in-depth analysis and valuable comments. Most of them helped us to further improve the quality of our manuscript. We followed the suggestions and revised the manuscript accordingly. A point-by-point response is given below. Comments by the referees are italic and in blue, our response is non-italic and in black. A pdf file with all changes marked in a track-change mode is also uploaded.

Thank you very much for all your efforts,
Lars Heinke and Mariana Kozłowska on behalf of all authors

Reviewer #1

The work of Mostaghimi and coworkers presents the computational design and analysis of various MOF with photoswitchable conductivity using the spiropyran photoswitchable ligand. Molecular design based on the interchromophore distance and using different substituent groups allows to tune the photoswitchable properties of the MOF to achieve several others of magnitude of difference in the charge carrier mobility between the two isomers. Throughout computational analysis based on DFT structure optimization, FF molecular dynamics and electronic coupling calculations is done for 8 MOF structures based on Cupaddlewheel units connected through the spiropyran photoswitch forming layers that are pillared connected with ligands of different sizes: dabco and bipyridine. The conclusions of the work are well supported by the simulations which are based on robust theory of charge transport in molecular materials and thermal effects introduced by real time MD. This work is certainly interesting for the materials chemistry community working in the understanding and molecular design of functional materials. There are however some points that need clarification in the manuscript.

#1. "In the abstract and introduction, it is mentioned that the trial and error approach is highly inconvenient in the design of MOF and photoswitchable materials. In this context it is not clear how do the authors choose the specific setup of copper paddlewheel metal nodes connected in layers by the photoswitch and pillared by an auxiliary ligand. Is this setup is based on previous work? Or there was some rationale for this setup?"

Answer:

MOFs are hybrid materials, where both metal nodes and organic linkers may be differently selected to tune the materials structure, topology, and functionality. The choice of metal nodes is still rather limited in comparison to millions of organic linker molecules. Therefore, our work and the sentence in the abstract were motivated by the trial and error approach of MOF design via linker selection.

For the MOFs in the manuscript, we use the copper paddle-wheel metal nodes based on our previous works (e.g. DOI:10.1021/ja1109826, DOI:10.1039/C0DT01818J, 10.1002/adma.201806324, 10.1016/j.micromeso.2015.03.018, 10.1038/srep00921). Such metal nodes are used in experiments due to the coordination geometry, crystallinity, low cost, and high thermal and chemical stability. In the decade-long expertise, we have found that Cu-based SURMOFs are especially stable and provide better control of the square planar coordination with various organic linkers. Due to our expertise in the field and the reproducibility of the SURMOF fabrication, we have focused our investigation on the design of linkers that can be incorporated in such MOFs.

The computational setup of this metal node was provided by the AuToGraFS software (DOI: 10.1021/jp507643v), where the library of various metal nodes, resulting in a diverse topology of the MOF, is available. AuToGraFS tools were implemented in our workflow, as depicted in Figure S1, and were used for building the initial MOF models for all our studies reported here, as well as previously (DOI: 10.3389/fmats.2022.840644). We have added the following explanation to SI:

“In the MOF-input WaNo, the layer and pillar linkers are uploaded in the xyz format, and the desired metal (i.e. metal node type/topology) is selected. Files with predefined and optimized metal topology, available in the AuToGraFS library, are used². For this study, we have use paddle-wheel Cu oxo-node called “Cu_pw6”. Either dabco or bipyridine were used as pillar linkers, coordinated to Cu oxo-node through the nitrogen atom (Cu...N); and either spiropyran or merocyanine (with different R-substitution) were used as layer linkers, i.e. coordinated to Cu through the oxygen atom from the carboxyl group (Cu...O).”

#2. “For the results shown in Figure 3 and the discussion in page 6, how the different distances between the SP and MC linkers is achieved in the DFT optimizations of Cu₂(SP)₂(dabco) and Cu₂(MC)₂(dabco) is not explained. In page 8 it is specified that the dabco ligand gives rise to 9.2 Å separation, and in page 9 the bipyridine ligand results in 13.7 Å separation. However, how the other distances are obtained is not clear.”

Answer:

Thank you for this remark. Cu₂(SP)₂(dabco) and Cu₂(MC)₂(dabco) were optimized in DFT. They were used for the visualization in Figure 3. Due to the length of the dabco linker, the distance between 2D sheets in MOFs (i.e. formed by the layer linkers), connected by the pillar, was 9.2 Å. The two neighboring layer linkers were extracted from the optimized structure using an in-house code “Achmol”. The code was published in the previous paper (DOI:[10.3389/fmats.2022.840644](https://doi.org/10.3389/fmats.2022.840644)). The extracted linkers were moved on a particular intermolecular distance using an automated script, and the electronic coupling was calculated for the molecules separated by the distance in the range of 8-15 Å. The distance screening was based on the center of mass distance between the two molecules in the dimer. No additional geometry optimizations of the dimer separated on diverse distances were performed.

Cu₂(SP)₂(bipy) and Cu₂(MC)₂(bipy) MOFs (explained further in the text) contain a different pillar linker, i.e. bipyridine, which has different length than dabco, therefore the distance between the linkers was 13.7 Å. We have not found any interactions between pillar linkers and layer linkers that affect the electronic properties of MOFs, therefore the change of the unit cell as a result of the pillar exchange was considered by shifting the layer linkers as explained above.

We have added the following description in the main body:

“Dimers of layer linkers (as depicted in Figure 3a-b) were extracted from the periodically optimized Cu₂(SP)₂(dabco) and Cu₂(MC)₂(dabco) MOF supercells using Achmol code¹⁷. Metal nodes were omitted, and the structures were neutralized by hydrogen atoms instead. Molecules in the dimer were moved on the specific distance, defined by center-of mass distance (COM), to mimic the length of the pillar linker (marked within green arrow in Figure 3). Electronic coupling matrix elements between linkers in such dimers, without any further geometry optimization, were calculated.”

We have added the following explanation in the Methods section:

“For the distance screening with the simultaneous electronic coupling calculations between layer linkers in the MOF two neighboring linkers were extracted from the optimized periodic MOF structure using Achmol and moved to a sequence of intermolecular distances, incrementing every 1 Å with an in-house script. The electronic coupling was calculated for linkers separated by a distance in the range of 8-15 Å. Lower distance values were not considered due to the overlap of atoms between the molecules, whereas at higher distance values the couplings are lower than 0.01 meV. A similar approach was used for the calculation of the electronic coupling between molecules extracted from MD snapshots (Figure S5).”

All changes are marked in a track-change mode.

#3. “In page 6, it is mentioned that “electronic coupling of linkers on the smaller distance was calculated using structures extracted from MD simulations”. Then at page 7 it is mentioned “Slightly better on-off photoswitching for linkers extracted from MD is found to be after OH-modification”, and at the end of the paragraph: “Electronic couplings mentioned were calculated only for one snapshot from MD, thus more sophisticated analysis is necessary.” Why only one snapshot for the OH case? Which snapshot was chosen and why? Are these MD at 300 K? How long are these dynamics?”

Answer:

We have not performed full DFT optimization of the MOF with OH-modified photoswitches for the distance-dependence of the electronic coupling. Analysis of MD data suggested that this type of MOF would be interesting for detailed inspection. Therefore, we have used the snapshots from MD at 300 K, performed with an identical setup as all other MD simulations, for the distance screening calculations. The extracted snapshot was selected based on the stability of the dimer formed in 10 ns MD simulations, i.e. we have checked the interaction pattern of the molecules and checked if such structures are formed several times in MD with a lifetime longer than other temporary structures. We understand that the former presentation of the results may be slightly misleading, therefore, we have deleted the explanation of the distance screening calculations of MOFs with OH-modified photoswitches.

More sophisticated analysis of the electronic coupling for MOFs with OH-modified linkers is now provided in the next paragraphs that follow, where the change of the electronic coupling for 250 selected snapshots (without an additional distance screening) is mentioned. We have also added the following text in the main body:

“Since it was previously demonstrated that vibrational flexibility of linkers in a MOF may plays an important role in the CT transport in a material,^{17,26} we conducted more detailed analysis of MD simulations. Specifically, simulations of eight MOFs: Cu₂(SP)₂(dabco), Cu₂(SP-OH)₂(dabco), Cu₂(SP-CHO)₂(dabco), Cu₂(SP)₂(bipy), Cu₂(MC)₂(dabco), Cu₂(MC-OH)₂(dabco), Cu₂(MC-CHO)₂(dabco) and Cu₂(MC)₂(bipy), are analyzed below. All calculations were performed using the same computational setup utilizing workflow depicted in Figure S1 with the flowchart explained in Figure S2.”

“DFT optimized (or pre-optimized) MOF structures were also used as an input for MD simulations of MOFs using Large-scale Atomic Molecular Massively Parallel Simulator⁹⁴ (LAMMPS) at 298 K (lammmps-interface and LAMMPS WaNo)⁹⁵. MD runs were performed using UFF4MOF^{96,97} force field. The 4×4×4 MOF supercells were simulated with a timestep of 1 fs. The equilibration was done using an NVE ensemble and a Langevin thermostat⁹⁸ for

0.3 ns, whereas the production MD runs were performed using the canonical (NVT) ensemble and Nosé-Hoover thermostat⁹⁹ for 10 ns.”

#4. “second paragraph in page 6 starts: “Since it was previously demonstrated that vibrational flexibility of linkers in a MOF may play an important role in the CT transport in a material, 10 ns MD simulations of Cu₂(SP)₂(dabco), Cu₂(SP-OH)₂(dabco), Cu₂(SP-CHO)₂(dabco), Cu₂(SP)₂(bipy), Cu₂(MC)₂(dabco), Cu₂(MC-OH)₂(dabco), Cu₂(MC-CHO)₂(dabco) and Cu₂(MC)₂(bipy) were performed for a simulated temperature of 300 K....”Do these include the previously discussed dynamics of Cu₂(SP)₂(dabco), Cu₂(SP-OH)₂(dabco), Cu₂(MC)₂(dabco) and Cu₂(MC-OH)₂(dabco)? Then later in the same paragraph: “In the following we analyze on-off photoswitching between Cu₂(SP)₂(dabco) and Cu₂(MC)₂(dabco)” Are these the same dynamics discussed before? Or different ones?”

Answer:

Yes, these are the same MD simulations. In the beginning, we focused on the change of electronic coupling as a function of the separation distance between the molecules in the MOF. For that, we have used DFT-optimized structures of unmodified photoswitches (see comment #2). Since it was previously demonstrated that vibrational flexibility of linkers in a MOF may play an important role in the CT transport in a material (DOI: 10.1039/D0SC07073D, 10.3389/fmats.2022), we have performed MD simulations of these MOFs and new candidates. We have not performed full DFT optimizations for modified MOFs.

During the calculation of the electronic coupling based on MD snapshots, we have found that OH-modified photoswitches show slightly different dependence, therefore, we decided to include distance screening calculations of electronic coupling also for these MOFs using MD snapshots, which were selected as described in #3. We understand that the explanation of the results may be slightly misleading, therefore, we have deleted the explanation of the distance screening calculations of MOFs with OH-modified photoswitches and explained the change of couplings from MD in the next paragraph:

“Since it was previously demonstrated that vibrational flexibility of linkers in a MOF may plays an important role in the CT transport in a material,^{17,26} we conducted more detailed analysis of MD simulations. Specifically, of simulations of eight MOFs: Cu₂(SP)₂(dabco), Cu₂(SP-OH)₂(dabco), Cu₂(SP-CHO)₂(dabco), Cu₂(SP)₂(bipy), Cu₂(MC)₂(dabco), Cu₂(MC-OH)₂(dabco), Cu₂(MC-CHO)₂(dabco) and Cu₂(MC)₂(bipy), are analyzed below. All calculations were performed using the same computational setup utilizing workflow depicted in Figure S1 with the flowchart explained in Figure S2. The dimer (and for the MC case also a trimer) of the layer linkers were extracted from the central part of the 4x4x4 supercell of MOF (see Figure 4). Visualization of the extracted pairs of molecules is depicted on the selected snapshots from the 10 ns MD runs in Figure S6-S13.”

All changes are marked in a track-change mode.

#5. “starting page 9: “To estimate the change in on-off SP□MC switching upon linker functionalization with OH and CHO groups, similar calculations were performed for Cu₂(SP-OH)₂(dabco), Cu₂(MC-OH)₂(dabco) and Cu₂(SP-CHO)₂(dabco), Cu₂(MC-CHO)₂(dabco), respectively. The dynamical evolution of the electronic coupling matrix elements is depicted in Figure S11-S12” Here again is not clear if these are different dynamics or it is referring to the ones previously mentioned in page 6?”

Answer:

The snapshots from MD simulations (that were used for the distance screening, explained below Figure 3 in the text) originate from the same MD simulations as used further to estimate the change of coupling in time (i.e. not as a function of the intermolecular distance). Therefore, the difference in the use of MD data is the following: for the distance screening, we extracted linkers from one (averaged) MOF snapshot, selected as described in #3, and moved them within the defined center-of-mass distance with the simultaneous coupling calculation. The coupling dependence on time was estimated using 250 MD snapshots without any additional distance screening. All MD simulations of 8 MOFs were performed with the same computational setup using automated workflow. We hope that changes introduced after the reviewer comment #4 will clarify the misunderstanding. We have added the following explanation in the main body:

“Since it was previously demonstrated that vibrational flexibility of linkers in a MOF may plays an important role in the CT transport in a material,^{17,26} we conducted more detailed analysis of MD simulations. Specifically, simulations of eight MOFs: Cu₂(SP)₂(dabco), Cu₂(SP-OH)₂(dabco), Cu₂(SP-CHO)₂(dabco), Cu₂(SP)₂(bipy), Cu₂(MC)₂(dabco), Cu₂(MC-OH)₂(dabco), Cu₂(MC-CHO)₂(dabco) and Cu₂(MC)₂(bipy), are analyzed below. All calculations were performed using the same computational setup utilizing workflow depicted in Figure S1 with the flowchart explained in Figure S2. The dimer (and for the MC case also a trimer) of the layer linkers were extracted from the central part of the 4x4x4 supercell of MOF (see Figure 4). Visualization of the extracted pairs of molecules is depicted on the selected snapshots from the 10 ns MD runs in Figure S6-S13. To demonstrate the time evolution of the electronic coupling between neighboring layer linkers, couplings between extracted linkers (similarly as depicted in Figure 3a-b, without any additional distance screening) in 250 snapshots from the last 5 ns of MD were calculated using MO approach (eq. 2-5) using DFT level of theory.”

#6. *“I am not sure if the last paragraph mentioning the experimental attempt to crystallize and characterize the proposed systems deserves 9 lines since, as the authors mention themselves, “Further experimental optimizations are essential and will be presented elsewhere.” Moreover, experimental method details are not given. The authors may consider to remove this part and shortly mention in the conclusions that experimental validation is ongoing.”*

Answer:

Thank you for this remark. By weighting this comment and the comment #2 of Referee 2 (who asks for more details), we decided to revise this paragraph giving the current status of the experimental progress:

“To experimentally reproduce the theoretical findings, the syntheses of surface-mounted MOF (SURMOF) films with the targeted structures (see Figure 2) were performed in a layer-by-layer fashion. With this established method, i.e. by subsequently immersing the substrates in the ethanolic solutions of the copper-acetate-metal-nodes and the linker molecules, SURMOFs with various pillared-layered structure have been prepared, see e.g. refs⁷⁵⁻⁷⁷. With the aim of synthesizing the MOF films with the designed structures, many parameters were varied and optimized, however, the obtained X-ray diffraction data show that the prepared films have no high crystallinity. Further experimental optimizations are essential and will be presented elsewhere.”

Reviewer #2

In this manuscript Heinke, Kozłowska and coworkers report on a modelled spiropyran-based Metal-Organic Framework. This are very nice results and a good fundamental study, which surely deserves publication in Commun. Chem.

There are however, some points that should be improved prior to acceptance:

#1. "Figure 1: Please increase the resolution. Moreover, Figure 1 closely resembles Figure4. of 10.1002/anie.201811458. The authors should be more careful about this!"

Answer:

Thank you. We followed the advice and modified Figure 1 accordingly.

#2. "I could follow the theoretical part of the manuscript and during that I was really looking forward to the experimental results. At the end of the manuscript, only a paragraph was reported on the experimental findings. Furthermore, it turned out that the theoretical prediction could not be reproduced, due to the lack of crystallinity..."

Finally, where are the experimental results? I was not able to find it, please include them. It seems to me that both groups were disappointed and simple did not add it. Why?"

Answer:

We tried to experimentally reproduce the theoretical findings. Unfortunately, although we put a lot of effort in it and we tried many different synthesis methods, we were not able to synthesize MOF films with the targeted structure and a high crystallinity. We added the text and stress this clearly in the manuscript:

"To experimentally reproduce the theoretical findings, the syntheses of surface-mounted MOF (SURMOF) films with the targeted structures (see Figure 2) were performed in a layer-by-layer fashion. With this established method, i.e. by subsequently immersing the substrates in the ethanolic solutions of the copper-acetate-metal-nodes and the linker molecules, SURMOFs with various pillared-layered structure have been prepared, see e.g. refs.60-62. With the aim of synthesizing the MOF films with the designed structures, many

parameters were varied and optimized, however, the obtained X-ray diffraction data show that the prepared films have no high crystallinity. Further experimental optimizations are essential and will be presented elsewhere.”

#3. “Some Typos:

- a. 79 the quote is before the point
- b. 145 the quote is before the point
- c. 186 were calculated”

Answer:

We have corrected these typos. Now it is: “...the charge injection.¹²”, “... were used.⁵⁶” and “The structures of all MOFs were calculated using...”

Reviewer #3

#1. “In the current manuscript, Mostaghimi et al. describe multi-scale simulations used to analyze the conduction switching in spiropyran-based MOFs. The topic is interesting and timely, but in my opinion the manuscript is not publishable in its current form. Besides some potentially rather serious general aspects and a rather superficial description of the methodology, my primary concern is that the model used to estimate electronic conductivities might very well not be suitable for the materials at hand due to their strongly varying degree of dynamic disorder.”

Answer:

Thank you for your comment, but we believe that our methodology covers all the relevant aspects for this system. When considering dynamic disorder, it is essential to distinguish between systems with and without delocalized polarons. In the present system all polarons are strongly localized on individual linkers due to the large intermolecular separation. In our previous study (DOI: 10.1039/D0SC07073D), we have also shown that structural fluctuations in MOFs break band-like transport and demonstrate a hopping dominated nature of the charge transport, including charge hopping processes. In such systems the only relevant source of dynamic disorder stems from the fluctuations in the coupling which are fully accounted for in our methodology.

We agree with the reviewer that in the case of delocalized polarons, a much more complex treatment of dynamic effects may be relevant, as for instance discussed for crystal-like and amorphous (i.e. disordered) materials (DOI: 10.1021/acs.accounts.1c00675, 10.1021/ja104380c, 10.1021/acs.jctc.6b00564), including works of Frattini et al. (e.g. DOI: 10.1002/adfm.201502386, 10.1038/nmat4970). Transient localization theory models aim mostly to describe charge transport on the border between delocalized and localized charge transport (DOI: 10.1021/acs.jpcc.8b11916). Indeed, they may be applicable for specific classes of MOFs, e.g. pentacene-based MOFs as reported in DOI: 10.1039/D0SC07073D, where delocalization has a chance to happen (between within two molecules only). However, only localized charges are possible in the MOF studied in our manuscript. It can be roughly estimated from the relation between the electronic coupling and the reorganization energy, as reported, for example in DOI: 10.1021/acs.accounts.1c00675 (Figure 4), i.e., $\xi = 2V/\lambda$, where V is the electronic coupling. If $\xi < 1$, a finite energy barrier between the two charge localized states (minima) exists, i.e. small polarons are formed. The reorganization energy (λ) of spiropyran (SP) is 0.35 eV and 0.46 eV for HOMO and LUMO

orbitals, respectively. The reorganization energies (λ) of merocyanine (MC) is 0.17 eV and 0.34 eV for HOMO and LUMO orbitals, respectively. (The reorganization energy λ , determined by the geometry changes of the molecule upon charging, was calculated using the four-point method with B3LYP/def2-TZVP.) Using the average electronic coupling values from MD (max. 0.03 eV, Table S1) or coupling calculated on DFT structures (Figure 3c,d) of max. 0.02 eV, we clearly see that max. value of $\xi = 0.35$ for the hole transport of MC and smaller than 0.17 for all other cases. This clearly indicates that delocalization of charge is hindered and, even if possible, it will only temporarily occur for some interaction patterns with high electronic coupling and only in the case of the hole transport of MC. This may be an additional indication of the on-off conduction switching observed. However, considering strong dynamical disorder (especially in the merocyanine case), this should not have an impact on the transport mechanism.

In the present paper, we aimed to show the change of the electronic coupling between linkers in the MOF, further indicating potential for on-off photoswitching and possible change of the photoconduction. We revised the manuscript for misunderstanding statements and we deleted calculated charge transfer rates that were calculated for several charge hops only (Table S1). All changes are marked in a track-change mode. Specifically, the following text was added to the main body:

“However, Marcus-type hopping transport represents only the localized CT transport, while other types of CT may also occur⁶⁹, e.g. band-like transport or CT on the border between delocalized and localized charge transport known as transient localization theory models^{60–62}. Still, the electronic coupling between the molecules in a material directly defines the propensity of its semiconducting properties, therefore, it may be used as an indication of the CT change upon the change of the chemical composition or topology of a material. In the present study, we do not focus on the prediction of intrinsic CT of the MOFs considered, but on the range of the electronic coupling change upon the photoconversion. “

“Considering the charge carrier mobility under hopping transport, which is often used to explain the semiconduction properties of MOFs due to rather large intermolecular distances^{28,66–68}, it is defined as:

$$\mu = \frac{eL^2}{k_B T} k_{CT}, \quad (6)$$

where L is the center-of-mass distance between linker molecules and k_{CT} is the CT rate defined, as an example, in eq. 1. Therefore, the CT rate and the charge carrier mobility, and thus the conductivity, depends quadratically on the electronic coupling, disregarding the reorganization energy, and deviations in the site energies. As such, the change in the electronic coupling can be used as an indication of the change in the conductivity of the system. Note that the final value of the charge carrier mobility depends on dynamic and energetic disorder in the system. The dynamic disorder is defined by vibrational flexibility of the material, leading to the change of the electronic coupling between molecules and its semiconducting characteristics. Impact of such motions, e.g. rotations, pedal motions, or dipole orientational disorder, involving changes in molecular conformations of assembled molecules, was reported for various classes of materials^{69–71}. Energetic (static) disorder is related to the spread or variation in the time-averaged energy levels of molecules comprising the material^{72,73}, which stems from the time-independent lack of perfect order. In our study, dynamic disorder of ordered MOFs refers to the local motions of linkers within the MOF structure. To account for it in the on-off conduction photoswitching, we calculated the electronic couplings between molecules extracted from the MD simulation at 298 K. Such an approach can directly demonstrate the deviation of the electronic coupling as a function of vibrational flexibility of the system. “

#2. “In the simulations the authors apparently assume that upon photoexcitation 100% of the chromophores switch between the spirocyan and the merocyanine form. At least this

would be my guess, as (like many other aspects) this seems to be never explicitly described in the manuscript. The authors ought to comment on whether this is a realistic assumption. In this context they also should consider that in the case of mostly one-dimensional charge transport along chromophore stacks (again I guess that this is what they assume here) a single “defect” (=non-switched molecule) can have a tremendous impact on the material’s charge-transport properties as the charge has hardly any chance to bypass that effect.”

Answer:

We considered the transformation of spiropyran-based linkers into merocyanine-based linkers in the MOF upon external stimulus, assuming 100% conversion as a model system to illustrate the behavior for the two ideal systems. We acknowledge such perfect conversion is unlikely to be achieved in practice. In our previous study on spiropyran embedded into the UiO-67 MOF pores (DOI: 10.1002/anie.201811458), a switching yield of approximately 70 % MC was achieved upon 365 nm UV light irradiation. However, it depends on the number of molecules in the pore (distance criteria) and the type of the MOF (polarization environment, DOI: 10.1021/acs.inorgchem.7b01908, 10.1002/cptc.201900193). Since we cannot provide valid information about the conversion of spiropyran in the MOFs studied, we investigated two different forms of MOFs, showing the overall possible effects in this idealized case. We agree with the reviewer that defects may impact effects observed too. They will lower the on-off ratio predicted in theory. We highlighted that the reported on-off ratios are potential ratios and the values reported should be treated as the highest possible. We have added information about these limitations in Conclusions:

“The study provides a deeper understanding of the photoswitching in SP-functionalized MOFs. It reports potential on-off ratios upon photoisomerization, however, several further effects may impact on the final conduction switching that were not considered in the present work, such as the isomerization yield between SP and MC and the presence of defects. At the same time, this study also provides guidelines which can be transferred to other switchable molecules like spirooxazine and diarylethene and to other classes of materials, in particular framework materials like covalent organic frameworks, where the factors mentioned may play a different role. In that way, it contributes to the fast development of smart materials.”

#3. “I wonder, whether the assumed structures are realistic – in particular, considering the rather long side-chains of the linkers, can the authors be sure that the conformation they studied is the only conceivable one? In view of the employed periodic boundary conditions it seems that they assumed that the side chains in every parallel linker points in roughly the same direction (an aspect that I again never found to be discussed explicitly). Can the authors exclude that the side-chains of consecutive linkers point in very different (maybe even opposite) directions. I suppose that this would completely change and presumably diminish charge transport, but would possibly not show up in MD simulations running only for a few (ten) nanoseconds when the initial configurations consists of essentially parallel side chains (i.e., there might not be enough time for the side-chains to flip).”

Answer:

Thank you for this consideration. MOF structures with the merocyanine-based linkers pointing out different directions are not possible. The distance between the linkers in one MOF unit (“pore”) is ca. 7-9 Å (see new Figure S3 and S4), while the length of the side chain is around 10 Å. Therefore, the switch of the side chain to the other direction is sterically forbidden. Secondly, such a switch (i.e. rotation) cannot happen, starting from the “opposite” conformation, because there is not enough space between MOF sheets to allow for such a move (MOFs with the dabco pillar will have max. 9 Å space). Even if we generate a MOF

with differently pointing outside chains (only in the case of spiropyran, because it is ca. 2 Å shorter), this should not significantly impact electronic couplings between SP molecules, because the HOMO orbital is localized mostly on the central ring (see Figure 6) that does not change the position upon different localization of the side chain. Indeed, such a case will impact electronic properties of the converted merocyanine MOFs since there will not be enough space for the photoisomerization to happen, lowering on-off photoswitching. We have found the reviewer concerns as an interesting point; therefore, we have added the following text in ms:

“The starting position of the layer linkers was selected in the way that the side chains point out in the same direction, as depicted in Figure S3 and S4. The opposite location of the side chains does not permit the formation of the merocyanine-based MOFs, because the intermolecular distances between the neighboring linkers are too small. If the opposite location of the spiropyran side chain happens during MOF fabrication, it will limit the photoconversion to the merocyanine form, impacting properties of a material. We exclude such effects in the present study.”

Figure S3. The visualization of the intermolecular distance between spiropyran-based layer linkers in the Cu₂(SP)₂(dabco) MOF: a) and c) distance between the closest atoms between two different linker pairs (MD snapshot at 5.7 ns). b) and d) time evolution of the intermolecular distance in MD at 298 K.

Figure S4. The visualization of the intermolecular distance between merocyanine-based layer linkers in the $\text{Cu}_2(\text{MC})_2(\text{dabco})$ MOF: a) and c) distance between the closest atoms between two different linker pairs (MD snapshot at 5.7 ns). b) and d) time evolution of the intermolecular distance in MD at 298 K.

#4. "The introduction is rather lengthy, more in the style of a review and not exactly to the point. In contrast, the "Results and Discussion" section is rather hard to follow. While I am not so much concerned about the lengthy introduction, a more accessible "Results and Discussion" section would be highly desirable."

Answer:

We have added explanations of the properties calculated and the results obtained. All changes are marked in track-change mode.

#5. "In the "Results and Discussion" section, a clearer explanation for why certain effects are observed would be desirable (for example, for the 10-fold increase of the coupling matrix mentioned in line 265)."

Answer:

We revised the manuscript and added clearer explanations. All changes are marked in a track-change mode.

#6. "In lines 343 ff the authors try to extract a connection between the energies of the frontier orbitals and the electronic coupling, while not providing a sufficiently clear explanation of the origin of that connection. Thus, I wonder, whether this connection is a pure coincidence, whether there are just correlations or, whether there is really a causality in the connection between frontier orbital energies and electronic couplings?"

Answer:

In general, there is a connection between electronic coupling and frontier orbital energies. In one of the methods to calculate electronic coupling based on Koopmans' theorem (DOI: 10.1063/1.2727480, 10.1039/C002337J), the transfer integrals are calculated as the half of the orbital energies of the dimer. This means that the electronic coupling between the HOMO orbitals of two molecules in a dimer is half of the energy difference between the HOMO and HOMO-1 orbital. Analogously, the coupling between the LUMO orbitals is half of the energy difference between the LUMO and LUMO+1 orbitals. This is an approximative way to calculate the electronic coupling that has been shown to give good results for cofacial dimers (DOI: 10.1063/1.1925611, 10.1039/B717752F). The exemplary orbital correlation diagram explaining the dimer approach for the electronic coupling calculation is depicted in Figure R1.

Figure R1. Orbital correlation diagram of the dimer approach for the electronic coupling calculation. The dimer level splitting $2|\beta|$ and $2|\beta^*|$ for the HOMO- and LUMO-derived dimer orbitals, respectively, corresponds to the coupling between orbitals (J is a half of $2|\beta|$ and $2|\beta^*|$ for J_{HOMO} and J_{LUMO} , respectively). LUMO becomes SOMO (single-occupied MO) when it is singly occupied. Orbital No 3 is the bonding combination of the two SOMO orbitals. According to this diagram, bonding in a dimer is formed as a result of overlapping the SOMO orbitals (single occupied molecular orbitals, depicted as single occupied LUMO). Then, $2|\beta^*|$ is the dimer level splitting, which reflects the overlap between the two LUMO orbitals. It is the measure of the charge transfer integral, i. e. electronic coupling element.

However, in our study we have found another correlation: the change in the value of the HOMO (or LUMO) energy between SP monomer and MC monomer corresponds to the on-off ratio, i.e. higher the difference between HOMO (or LUMO) orbital of spiropyran and merocyanine is, the stronger is on-off ratio (the change in the coupling obtained from the calculation of the dimer). To our knowledge there is no direct way to correlate this observation to the electronic coupling. Even if orbital energy splitting defines the electronic coupling values, the relation to the actual value of a HOMO (or LUMO) of spiropyran and merocyanine and on-off effect, i.e. low coupling/high coupling, is unclear. More screening calculations and experiments are necessary to validate the correlation; however, this is an observation we have seen in the current study. We have added an explanation in the manuscript:

“As to our knowledge, there is no direct explanation of the change in the HOMO (or LUMO) value of two photoisomers and the change of the electronic coupling between their homodimers. Indeed, in the Koopmans' method for the coupling calculation⁶³ the energy splitting between HOMO and HOMO-1 (or LUMO and LUMO+1) orbitals is used to estimate the strength of the transfer integral, however it is not directly connected to the orbital differences between different isomers.”

All changes are marked in a track-change mode.

#7. “The analysis of charge transport is based on Marcus-type hopping rates. Considering the tiny electronic couplings between the different chromophores, I suppose assuming pure hopping transport is fair enough. What, however, could be highly problematic is that for calculating the charge transfer rates, the authors appear to use the average electronic couplings extracted from dimers found in the MD trajectories. To my understanding, just taking the average value and completely disregarding the degree of variation of the transfer rates is inappropriate and will produce even qualitatively incorrect trends for systems like the present ones in which not only the average value of the coupling, but also its degree of variation appears to change significantly from system to system. “

Answer:

We have added charge transfer rates as an illustration and calculated rates based on the average and the highest coupling values (old Table S1). We agree with the reviewer that such a definition of rates does not show the degree of variation of the electronic coupling on the CT rate. To avoid misunderstanding, we deleted previously calculated rates from Table S1 and we focus on the variation of the electronic coupling as a function of the photoisomer in the MOF and the possible on-off photoswitching. We investigated 250 snapshots from the last 5 ns simulation, i.e. one snapshot every 20 ps and we observed a substantial fluctuation. However, if we consider the system over a shorter amount of time, e.g. up to 0.5 ns (see Figure 5a), we can observe that couplings fluctuate less. Considering the typical charge transfer rates in MOFs in the range of 10-100 picoseconds (DOI: 10.1021/jacs.9b08078, 10.1021/acs.jpca.8b01192, 10.1021/jacs.7b13211), such temporary (stronger) couplings may be still enough for the charge to hop and propagate. The following explanation was added to the main body:

“Therefore, in the present study we focus on the change of the electronic coupling (i.e. transfer integral) between linkers in the MOF upon the photoconversion, which indicates the degree of possible on-off conduction photoswitching. Electronic coupling is the microscopic property, describing the electronic interaction between two molecules or fragments, that leads to the transfer or sharing of electrons. It is connected to the probability of electron transfer between two molecules; therefore, it is often used in charge transfer (CT) theories, e.g. in the Marcus theory of hopping transport.”

“However, Marcus-type hopping transport represents only the localized CT transport, while other types of CT may also occur⁵⁹, e.g. band-like transport or CT on the border between delocalized and localized charge transport known as transient localization theory models⁶⁰⁻⁶². Still, the electronic coupling between the molecules in a material directly defines the propensity of its semiconducting properties, therefore, it may be used as an indication of the CT change upon the change of the chemical composition or topology of a material. In the present study, we do not focus on the prediction of intrinsic CT of the MOFs considered, but on the range of the electronic coupling change upon the photoconversion.”

“We have to point out that in both cases, i.e. for HOMO and LUMO orbitals of MC, the electronic coupling is strongly varying, which would impact the overall CT in a MOF, increasing possible charge localization. At the same time, the temporary high MO overlaps permit possible charge transfer, however, the transport would be of high anisotropic and dynamic nature.”

#8. *“What is actually quite confusing is this context is that in the method section the authors cite Refs. 65, 66, and 67 as papers on which their calculation of electronic couplings is based. In these papers the carrier mobility is explicitly correlated with the energetic disorder, which according to equations (1) and (2) is not done here! In this context, I wonder, whether adopting the simple model to account for energetic disorder used in the above references would actually suffice for the highly anisotropic transport present in the studied systems. “*

Answer:

We have mentioned Refs. 65, 66, and 67 to explain the methodology based Quantum Patch method which is a general quantum embedding method but was used for densely packed amorphous organic materials, which have strong disorder. Here, we used this package only to calculate the electronic couplings. Indeed, the method was fully described in reference DOI: 10.1021/ct500418f. We understood that linking the methodology to all other references is misleading, therefore, we deleted other references.

We do not calculate energetic disorder in our manuscript, because such disorder for ordered systems as MOFs is negligible. In the references mentioned above amorphous materials and organic crystals were studied. Assembly in MOFs is different and the intermolecular distances between molecules are higher. Moreover, conformational movements of organic linkers, that are restricted in the MOF scaffold by coordination to the metal nodes, differ significantly. Therefore, coupling values, dynamical disorder, energetical disorder, possible charge hopping pathways etc. will be different, and simple models are not applicable. *We have added the following explanation in the main body:*

“In our study, dynamic disorder of ordered MOFs refers to the local motions of linkers within the MOF structure. To account for it in the on-off conduction photoswitching, we calculated the electronic couplings between molecules extracted from the MD simulation at 298 K. Such an approach can directly demonstrate the deviation of the electronic coupling as a function of vibrational flexibility of the system.”

#9. *“To consider also anisotropies, the authors might want to take a closer look at the dynamic disorder model developed over the years by Fratini et al.. In fact, some years ago Troisi, Fratini et al. published a paper in Nature Materials in which they not only eluded to the crucial (negative) implications of highly anisotropic transport, but in which at some point they also claimed that the magnitude of the transfer integral becomes irrelevant for charge transport in organic semiconductors in a dynamic disorder scenario. If one takes a careful look at the Supporting Information of that paper, one realizes that this statement is apparently true, when the absolute magnitude of the variation of the electronic coupling is proportional to the value of the electronic coupling. What this shows is that the degree to which the transfer integral varies over time is as important for the transport properties as the average value of the electronic coupling. Therefore, in cases in which not only the electronic coupling changes between different systems but also its variation over time it is simply not sufficient to only consider the average value.*

To cut a long story short, I have serious concerns that analyzing trends in charge transport in the systems considered here using equations (1) and (2) is strongly misleading and more sophisticated models (possibly even going between what has been described in Refs. 65-67) need to be employed.”

Answer:

We understand the concerns of the reviewer. We fully agree that works of Troisi, Fratini, et al. advance the current state-of-the-art and show an impact of the coupling fluctuations on charge transport. As pointed out in previous comments, the reported charge transfer mechanisms were developed and validated on (strongly) coupled crystals of organic molecules, where charges are delocalized. Therefore, they introduced the transient localization of the carriers' eigenstates, which is a result of off-diagonal dynamic disorder and is related to the large fluctuation of intermolecular transfer integrals (DOI:10.1002/adfm.201502386, 10.1021/acs.jpcc.8b11916). In our system, charges are localized due to the relatively high separation of the molecules. There is (fortunately) no reason to believe that the dynamic disorder effects discussed in the papers mentioned above are relevant here. Therefore, we use classical molecular dynamics to extract snapshots mimicking movements of organic molecules assembled in MOF. This means we calculate the change of the electronic coupling (and its average) from the classical movement of molecules. With this, we aimed to demonstrate vibrational fluctuations of molecules and the change of their electronic coupling explicitly. However, we think that it is important to refer the reader to the investigations mentioned and we have cited them in the text. We have added the following statement in the main body:

“However, Marcus-type hopping transport represents only the localized CT transport, while other types of CT may also occur⁶⁹, e.g. band-like transport or CT on the border between delocalized and localized charge transport known as transient localization theory models⁶⁰⁻⁶². Still, the electronic coupling between the molecules in a material directly defines the propensity of its semiconducting properties, therefore, it may be used as an indication of the CT change upon the change of the chemical composition or topology of a material. In the present study, we do not focus on the prediction of intrinsic CT of the MOFs considered, but on the range of the electronic coupling change upon the photoconversion.”

We use Marcus-type CT, because “Considering the charge carrier mobility under hopping transport, which is often used to explain the semiconduction properties of MOFs due to rather large intermolecular distances^{28,66-68}”, which is now added in the main body.

“In our study, dynamic disorder of ordered MOFs refers to the local motions of linkers within the MOF structure. To account for it in the on-off conduction photoswitching, we calculated the electronic couplings between molecules extracted from the MD simulation at 298 K. Such an approach can directly demonstrate the deviation of the electronic coupling as a function of vibrational flexibility of the system.”

For more detailed explanation please refer to the reply on the reviewer's comment #1.

#10. For calculating Figure 3, what is the actual structure of the model system that the authors use to increase the center of mass distance between the chromophores? Do they only consider the linkers, disregarding the metal nodes and the “pillars”? Do they then only consider linker dimers and if so do they leave their structures like in the bulk system at the equilibrium distance or do they optimize the atomic positions for every distance? In fact, I would exclude that in the actual systems the structures stayed the same, if one increased the distance.”

Answer:

Thank you for this remark. Dimers of layer linkers (as depicted in Figure 3a-b) were extracted from the optimized periodic MOF supercell using Achmol code (published DOI:

10.3389/fmats.2022.840644). We focus on the change of the electronic coupling between spiropyran- and merocyanine-based MOFs, therefore, pillar linkers and metal nodes were skipped. In our previous study (DOI: 10.1039/D0SC07073D), we have shown that metal nodes do not significantly impact such transport.

Molecules in the dimer were moved on the specific distance, defined by center-of-mass distance (COM), to mimic the length of the pillar linker (marked in green in Figure 3). Electronic coupling between linkers in such dimers, without any further geometry optimization, were calculated. We agree with the reviewer that geometry optimization with linkers separated on various distances may be slightly different, however it will not improve results, while the whole MOF scaffold, introducing structural confinement to the linkers, will not be present. We have added the respective explanation to the text:

“Dimers of layer linkers (as depicted in Figure 3a-b) were extracted from the periodically optimized $\text{Cu}_2(\text{SP})_2(\text{dabco})$ and $\text{Cu}_2(\text{MC})_2(\text{dabco})$ MOF supercells using Achmol code¹⁷. Metal nodes were omitted, and the structures were neutralized by hydrogen atoms instead. Molecules in the dimer were moved on the specific distance, defined by center-of mass distance (COM), to mimic the length of the pillar linker (marked within green arrow in Figure 3). Electronic coupling matrix elements between linkers in such dimers, without any further geometry optimization, were calculated.”

All changes are marked in a track-change mode.

#11. “No details are provided on how dimer structures were extracted from molecular dynamics runs. Did the authors only extract dimers stacked on top of each other and under which circumstances were trimer structures (like those shown in Figure 4b) extracted?”

Answer:

We have used 4x4x4 supercell for MD simulations. Dimers were extracted specifically selecting those located in the central region to minimize any potential periodic boundary effects. In addition, we wanted to include the impact of the neighboring molecules in the most adequate way. In the case of the unmodified merocyanine MOF (Figure 4b), we have observed the formation of stacking channels. Therefore, a trimer was extracted in addition and the dimer. For a trimer, the coupling between molecules “1” and “2”, “2” and “3”, “1” and “3” (three different dimers) was calculated and used to predict the superexchange-type coupling. The selection of both was always performed considering molecules in the central part (see Figure R2).

Figure R2. Visualization of the merocyanine linkers within $\text{Cu}_2(\text{MC})_2(\text{dabco})$ MOF, considered for the calculation of superexchange-type coupling (trimer) and direct coupling (dimer).

We have added the explanation in SI:

“MD trajectories were saved every 1000 steps, which resulted in 10300 frames. 1030 frames were analyzed using VMD software⁸. Equilibration data (first 30 frames) were not taken for further analysis. The detailed analysis of MD trajectories was necessary to enumerate the pair of layer linkers, representing a dimer used for further calculations (depicted in Figure 4). Since the enumeration of residues (defined via force field) is random, the pair of monomers had to be searched for all individual MOFs studied. 250 snapshots (every 20 ps) of the selected dimer (and trimer, in the case of Cu₂(MC)₂(dabco)) from the last 5 ns MD simulations were extracted using in-house script. They were later automatically hydrogenated (COO- groups that were linked to Cu metal nodes in a MOF) using Atomic Simulation Environment (ASE)⁹ and openbabel¹⁰. For each of the snapshots, the electronic coupling between the molecules was calculated using equations 2-5 in the main body using Quantum Patch¹¹.”

#12. “The use of the term “superexchange” is somewhat misleading. This term is borrowed from exchange coupling in ferromagnets, when the spins of magnetic metal ions are coupled e.g. via non-magnetic oxygen ions. Thus, even if the term superexchange has been used before in the context of charge transport, it is kind of a misnomer, but what is more relevant is, what the authors actually mean by it: For a structure like the one shown at the top panel of Figure 4b, do they mean that a charge carrier does two independent hops between neighboring Pi-systems (where I would not see any relation to superexchange) or does it directly hop from the leftmost to the rightmost Pi-system with the central ring only acting as “mediator” (which one might consider as a Coulomb-coupling equivalent to superexchange).”

Answer:

With the superexchange we have meant the second case described by the reviewer, i.e. charge hopping between two molecules via the third one, which is playing a role of the mediator. We have calculated such couplings using the same scheme as was reported in DOI: 10.1021/acsnano.6b03226, mentioned in the manuscript. We have added the explanation in the main body:

“In this mechanism, the electronic coupling between two linkers is enhanced by the third linker, playing a role of a mediator. In this sense, the coupling between layer linkers, located on the parallel planes (i.e. in a dimer that we consider in this work) is better, when the linker from another axis mediates the transport. For that, three coupling values are calculated, i.e. between all possible dimer structures forming a trimer, and the superexchange coupling is obtained using equation 7²⁸:

$$J_{gg'}^{tot} = J_{gg'}^{dir} + J_{gg'}^{sx} = J_{gg'}^{dir} + \sum_h \frac{J_{gh}^{dir} \times J_{hg'}^{dir}}{\Delta E_{sx} + \frac{1}{2}\lambda_h}, \quad (7)$$

where λ is the reorganization energy, $J_{gg'}^{dir}$ is the direct electronic coupling from site g to g' , $J_{gg'}^{sx}$ is a coupling matrix element of a superexchange process (i.e. charge hop from g to g' via h), h is the virtually occupied (intermediate, mediating) host state.

The superexchange part of the coupling would then depend on the direct coupling between g to h and h to g' and reorganization energy. The denominator ΔE_{sx} defined as:

$$\Delta E_{sx} = E_h - \frac{1}{2}(E_g + E_{g'}), \quad (8)$$

indicating the difference in the site energies of the virtual (host) state, is skipped in the present study since the mediator molecule is of the same type as the molecules involved in the charge hop.”

All changes are marked in a track-change mode.

#13. “In the methods section, the authors mention the SimStack workflow (also depicted in Figure S1). How does workflow correlate with the description of the methodology in the lines after line 406. Is the procedure described there performed after the SimStack simulations, is it part of it. A much more clear and consistent description would be highly desirable.”

Answer:

The SimStack workflow was used to generate all MOF structures, optimize or pre-optimize them in the plane wave DFT (in VASP) and simulate at 300K using MD (in LAMMPS). All steps are depicted in Figure S1. In the Methods section we described the concrete setup that was used in the WaNos of the SimStack to perform calculations. MD trajectories were analyzed separately and snapshots were extracted using automated codes. They were later coupled to Quantum Patch to calculate electronic couplings. The Quantum Patch algorithm is available within SimStack too, however, in the present study it was not directly used in one workflow. Therefore, the separation symbol is marked in Figure S1. Ideally, we aim to use the whole procedure as one workflow, but molecular fragments (residues) are randomly assigned by the LAMMPS interface and we could not automate the dimer selection and snapshot extraction steps. We added a clearer explanation in the Methods section and SI.

#14. “In fact, the entire modelling procedure in the methods section is described not very clearly. For the sake of reproducibility, a more coherent description of what has actually been done under which circumstances and to obtain which quantities would be needed.”

Answer:

We have revised the Methods section and added more detailed description of the methodology in SI.

#15. “There are many ways to calculate the electronic coupling between frontier orbitals and there is a bunch of literature on pros and cons of the different approaches. Considering the high importance of the electronic couplings for the present manuscript, the authors should provide a consistent description of which of them they actually chose (merely citing an approach developed to calculate charge carrier mobilities in general is not sufficient for understanding of how, in detail, the electronic couplings have been calculated).”

Answer:

In our study, the electronic coupling was calculated as direct overlap between the orbitals using Löwdin orthogonalization with Fock- and overlap matrices explained now in equations 2-5. The approach is described in DOI: 10.1063/1.2727480, 10.1016/S0065-3276(08)60339-1 and 10.1103/PhysRevB.83.155208. For this, molecular orbitals obtained

from a calculation on the isolated linker (i.e. monomer) was used as the basis set for the calculation on the dimer system, i.e. two linkers from the MOF. For our calculations, we employed the Quantum Patch software. The protocol is also available within SimStack, therefore allows for the direct computation of couplings in structures obtained in previous computational steps of the workflow (see Figure S1). We have added the respective description in the main body. All changes are marked in a track-change mode:

“Mathematically, the transfer integral, J , is represented as follows :

$$J_{if} = \langle \varphi_i | \hat{H} | \varphi_f \rangle, \quad (2)$$

where, φ_i and φ_f are the wavefunctions of the initial and final states, and \hat{H} is the Hamiltonian operator representing the total energy of the system. Thus, the electronic coupling as such is a matrix element, providing a measure of the overlap between the initial and final state wavefunctions and the energy involved in the transition. It is known to be estimated using several computational techniques that can be based on different levels of theory. Among the most commonly known techniques are methods based on Koopmans’ theorem, molecular orbitals (MO) and within perturbation theory⁶³. Here, we have applied a molecular orbitals approach to calculate the electronic coupling between spiropyran linkers in SP-based MOFs and merocyanine linkers in MC-based MOFs. For this, MO obtained from a calculation on the isolated linker (i.e. monomer) was used as the basis set for the calculation on the dimer system, i.e. two linkers from the MOF. Then the electronic coupling was calculated with a Löwdin orthogonalization method^{64,65} using Fock- and overlap matrices of the dimer:

$$J_{ij} = \frac{(H_{if} - \frac{1}{2}(H_{ii} + H_{ff})S_{if})}{(1 - S_{if}^2)} \quad (3)$$

with

$$H_{if} = \langle \varphi_i | \hat{H}_{KS} | \varphi_f \rangle \quad (4)$$

and

$$S_{if} = \langle \varphi_i | \varphi_f \rangle. \quad (5)$$

For hole (electron) transport, φ_i and φ_f are the HOMO and the LUMO molecular orbitals of the respective isolated monomers (linkers) and H_{if} is the charge transfer integral. \hat{H}_{KS} is the Kohn-Sham operator of the neutral dimer of two molecules, whereas H_{ii} and H_{jj} are the site energies of the two monomers. S_{if} is the spatial overlap. “

#16. “Why did the authors switch between functionals and between van der Waals corrections in the different steps of their simulations. This appears inconsistent and can be done only, if a proper justification is provided.”

Answer:

We have used PBE functional for the optimization of MOFs in the plane wave DFT due to the extremely high computational cost of similar calculations with hybrid B3LYP. At the same time, we use B3LYP for all other calculations since it is known to result in lower errors than PBE (DOI: 10.1063/1.1390175, 10.1002/ange.202205735). We have used the Tkatchenko-Scheffler method with iterative Hirshfeld partitioning as the dispersion correction (formally identical to the DFT-D2 method), because it considers variations in the vdW contributions of atoms due to their local chemical environment. It was applied for the full geometry optimization of Cu₂(SP)₂(dabco) and Cu₂(MC)₂(dabco) pillared MOFs.

At the same time, we have noticed that results with Grimme D2 dispersion correction are similar, but computationally cheaper; therefore, we have used that for the pre-optimization of other MOFs. Moreover, such an approach was recommended for other flexible nanoporous MOFs (DOI: 10.1063/1.5030493).

#17. “How large were the supercells assumed in the MD runs and did the authors observe any impact of the supercell-size on their results? How did they heat their structures? What thermostats were used? etc”

Answer:

The simulations were performed using 4×4×4 supercells of MOFs. This indicates a total of 192 linker molecules in the simulation box ($\frac{2}{3}$ of layered linkers and $\frac{1}{3}$ pillar linkers). With this we aimed to avoid any periodic boundary effects and represent the dynamics of centered linkers with neighboring molecules as a realistic MOF confinement. We do not expect any change in the motion of linkers, if we consider larger boxes.

The equilibration was done using an NVE ensemble and a Langevin thermostat for 0.3 nanoseconds. The production MD runs were performed using the canonical (NVT) ensemble and Nosé-Hoover thermostat. No heating procedure was applied, however, the position of metal ions was recentered to their center of mass in all directions. We have not observed any structural distortions or atom overlaps. We have added this information in the Methods section and SI:

“The 4×4×4 MOF supercells were simulated with a timestep of 1 fs. The equilibration was done using an NVE ensemble and a Langevin thermostat⁹⁵ for 0.3 ns, whereas the production MD runs were performed using the canonical (NVT) ensemble and Nosé-Hoover thermostat⁹⁶ for 10 ns. The position of metal ions was re-centered and constrained to their center of mass in all directions based on the DFT data. All other parameters were taken as specified in lammmps-interface⁹², e.g. real unit with LJ and Coulomb interactions cut-off 12.5 Å, as well as the use of thermostats for MOFs.”

. All changes are marked in the track-changes mode.

#18. “Regarding the data availability, I understand that journals do not yet insist on all calculations being uploaded to dedicated databases. Still with such databases readily available, I would consider it as “good scientific practice” to make simulation data of scientific papers publicly available”

Answer:

Thank you.

We have uploaded data into the NOMAD repository

<https://nomad.int.kit.edu/nomad-oasis/gui/user/uploads/upload/id/tMKbbMH0SEO8wk1eXJhDJg>

However, we need to have a DOI for the article to proceed with the final publication of the data. We cannot edit the entry in NOMAD, if we accept final data storage now. However, we will do that upon the finalization of the manuscript. We would appreciate it if the reviewer would allow us first to know about the decision about the manuscript. Thank you for understanding.

Reviewers' comments:

Reviewer #1 (Remarks to the Author):

The authors have made a great effort to answer all reviewers' comments and have clarified all points in the revised manuscript. Further review is not needed. I recommend publication.

Reviewer #3 (Remarks to the Author):

The authors have clearly improved the manuscript, addressing the points raised in my original report in one way or the other. Some issues like assuming a switching ratio of 100% or neglecting different linker orientations that would inhibit the switching makes the studied system not overly realistic, but explicitly mentioning these shortcomings of the study is certainly preferable over simply ignoring them. Also attempting to calculate charge carrier mobilities from the temporarily varying transfer integrals would have been appealing, but the chosen strategy of no longer claiming to calculate mobilities but rather focusing on (individual) transfer rates is acceptable, bearing in mind that an actual calculation of mobilities would be beyond the scope of the paper. Therefore, I consider the manuscript fit for publishing provided that the authors address the remaining issues mentioned in the following:

major:

- In line 311 the authors mention averaged structures from MD simulations. How were they determined and what is their significance?

- line 317: what is an "MD-averaged snapshot"?

- I am afraid that Figure 4 is still very confusing. Panel a: Is the dimer shown in the inset formed from linkers in two consecutive layers of which only one is actually shown in the main plot? While one still might understand that, it is not at all clear which other linker would be the "origin" of the central element/ring in the center of the trimer in panel b. Maybe a different viewing direction and showing more than one layer of the MOF might help to understand that?

- line 372: the authors quantify an increase of the electronic coupling by a factor of ~87-100. How exactly is this ratio defined?

- line 393: I do not see the grey marking in Figure 4 mentioned here.

- page 10: as already mentioned in the original report, superexchange in the present context is a misnomer as it is intimately related to exchange interactions, while the present manuscript is concerned with a Coulomb couplings. Therefore, a term like "superexchange-like" would seem more appropriate.

- What exactly is the meaning of Figure 5b? Is it simply the second term in equation (7), such that the total transfer rate would be the sum of the values plotted in panels a and b? If so, do the rather frequently observed high coupling values in Figure 5b mean, that the "mediating" group from the neighboring linker very often fills the space between the side groups of two linkers in consecutive layers?

- In line 519, the authors state that they find a "four orders of magnitude improved conduction". As the conductivity is no longer explicitly calculated, this should be reworded.

minor:

- I think it would help to more clearly discuss the limitations of the major assumptions of a 100% switching rate and a misalignment of the linkers.

- The authors might mention in the context of equation (6) that kCT between neighboring linkers varies by orders of magnitude over time and that it is far from straightforward to determine an effective kCT that one could directly insert into the equation. Still their time-resolved calculation of kCT reveals clear trends.
- The entire discussion on page 11 is based on data that are only shown in the Supporting Information. Thus, it is very difficult to follow that discussion. I wonder, whether including at least the most relevant of those data also into the main manuscript would be beneficial.
- The discussion in lines 477-480 has been added in response to a comment of an issue with the original paper mentioned in my first report. As such, it has been a useful statement in the referee reply. I am, however, not sure, whether including it in its current form into the main manuscript is really useful, as there it is quite out of context and would probably just confuse the reader.
- sometimes the construction of the sentences is very “involved”.

Dear referees,

We thank you for your time and effort in reviewing our manuscript COMMSCHEM-23-0153A. We are thankful for the appreciation of our work and for in-depth analyses and valuable comments.

We followed the suggestions and revised the manuscript accordingly.

A point-by-point response is given below. Comments by the referees are italic and in blue, our response is non-italic and in black, the changes in the manuscript are black, italic and stressed by quotation marks. In the manuscript, the changes are highlighted in yellow.

A pdf file with all changes marked in a track-change mode is also uploaded.

Thank you very much for all your efforts,

Lars Heinke and Mariana Kozłowska on behalf of all authors

Reviewer #1

The authors have made a great effort to answer all reviewers' comments and have clarified all points in the revised manuscript. Further review is not needed. I recommend publication.

Answer:

We thank the referee for the appreciation of our work and for the recommendation to publish the manuscript.

Reviewer #2

The authors have clearly improved the manuscript, addressing the points raised in my original report in one way or the other. Some issues like assuming a switching ratio of 100% or neglecting different linker orientations that would inhibit the switching makes the studied system not overly realistic, but explicitly mentioning these shortcomings of the study is certainly preferable over simply ignoring them. Also attempting to calculate charge carrier mobilities from the temporarily varying transfer integrals would have been appealing, but the chosen strategy of no longer claiming to calculate mobilities but rather focusing on (individual) transfer rates is acceptable, bearing in mind that an actual calculation of mobilities would be beyond the scope of the paper. Therefore, I consider the manuscript fit for publishing provided that the authors address the remaining issues mentioned in the following:

Answer:

We thank the referee for the appreciation of our work and the careful evaluation that improves the quality of our manuscript. We have clarified the remaining issues in the point-by-point replies below.

major:

- In line 311 the authors mention averaged structures from MD simulations. How were they determined and what is their significance?

Answer:

After 10 ns MD simulation of SP- or MC-based MOFs and the calculation of the electronic couplings in the 250 extracted snapshots, we have used the structure with the highest electronic coupling for further distance screening calculations. Since such structures were extracted in most cases after 6 ns MD runs, we were sure that they were representing

structures far away from the starting conditions or equilibration, acquiring some averaged state at 300K. Since such statement may be misleading, we have corrected the sentence in line 311 to:

“As a result, electronic coupling of linkers on the smaller distance was calculated using structures extracted from MD simulations (see Figure S5a) at 6.39 ns and 7.61 ns for SP and MC, respectively, where linkers show the strongest coupling upon the structural relaxation at room temperature. In this case, the vibrational flexibility of linkers permitted the change of the chromene unit orientation and lower screening distance.”

- line 317: what is an “MD-averaged snapshot?”

Answer:

As explained in the previous comment, we corrected the sentence in the revised manuscript to:

“Slightly lower absolute values of couplings and ratios were obtained from the distance screening calculations using the same MD-simulation extracted MOF structures (Figure S5b).”

- I am afraid that Figure 4 is still very confusing. Panel a: Is the dimer shown in the inset formed from linkers in two consecutive layers of which only one is actually shown in the main plot? While one still might understand that, it is not at all clear which other linker would be the “origin” of the central element/ring in the center of the trimer in panel b. Maybe a different viewing direction and showing more than one layer of the MOF might help to understand that?

Answer:

Thank you. We have followed the suggestions of the reviewer. The new version of Figure 4 (as shown below) is added in the revised paper.

- line 372: the authors quantify an increase of the electronic coupling by a factor of ~87-100. How exactly is this ratio defined?

Answer:

The coupling between the SP linkers (HOMO) from the MD-simulation snapshots is around 1.00-1.15 meV. On the other hand, it is approximately 100 meV for MC (HOMO) with some higher fluctuations (see Figure 5). So, the ratio is 87 and higher. We have explained these values in the description of Figure 5 in the main body, in the text below Figure 4. We assume that these data may be confusing when comparing couplings in Table S1, where the average coupling of MC (26.8 meV) and SP (1.15 meV) is given. To make the analysis more quantitative, in the revised paper we use the average values of the electronic coupling over 5 ns to estimate the on-off ratios. Due to the significant fluctuations of the coupling over time, we provide also the standard deviation, as well as the geometric average with the geometric standard deviation. The following text is now added in the manuscript:

“The difference in the vibrational flexibility of the isomers and the difference in their electronic properties permits an increase of the electronic coupling between the HOMO orbitals of 23 times during the on-off SP⇌MC-switching, considering the average values over 5 ns. Here, the arithmetic mean of coupling for MC and SP is 26.80 ± 27.77 meV and 1.15 ± 2.03 meV (see Table S1), while the geometric mean is 15.27 meV and 0.55 meV with the geometric standard deviation of 3.75 and 3.56, respectively. Due to the significant fluctuations of the electronic coupling over time at 300 K, there is no straightforward way to calculate the effective CT rate (eq. 1) using the average value of the coupling. Therefore, the approximated CT rate may possibly increase by 530 or even higher (see the multitude of couplings for MC over 50 meV in the upper panel in Figure 5a), when the separation distance between the linkers is around 9 Å (corresponding to a dabco pillar).”

- line 393: I do not see the grey marking in Figure 4 mentioned here.

Answer:

Thank you for this notice. The gray marking is now added in Figure 4.

- page 10: as already mentioned in the original report, superexchange in the present context is a misnomer as it is intimately related to exchange interactions, while the present manuscript is concerned with a Coulomb couplings. Therefore, a term like “superexchange-like” would seem more appropriate.

Answer:

We have followed the reviewer suggestion and use the term “superexchange-like” instead of “superexchange”.

- What exactly is the meaning of Figure 5b? Is it simply the second term in equation (7), such that the total transfer rate would be the sum of the values plotted in panels a and b? If so, do the rather frequently observed high coupling values in Figure 5b mean, that the “mediating” group from the neighboring linker very often fills the space between the side groups of two linkers in consecutive layers?

Answer:

Figure 5b represents not only the second term in equation 7, but the whole electronic coupling between linkers from the neighboring layers with the coupling originating from the superexchange-like processes. To clarify this better, we have improved the explanation in the text to:

“From Figure 5b, we see that the electronic coupling between LUMO orbitals of MC, which includes the impact of the superexchange-like process (through space) in $\text{Cu}_2(\text{MC})_2(\text{dabco})$ is lower than direct hopping between MC units in the MOF (depicted in

blue in the lower panel in Figure 5a). However, the coupling via the superexchange-like mechanism (see eq. 7) between HOMO orbitals is still relevant:"

Couplings reported for MC in Figure 5a and 5b do not originate from identical dimers. Since the MOF calculated in MD comprises 128 MC linkers, each representing independent structural moves, different dimers of layer linkers are formed (for example depicted in Figure S7). During the analysis of MD trajectories, we have noticed not only the formation of "traditional" dimers, but also the pi-stabilized channels, where superexchange-like processes are involved in the electronic coupling between linkers. In order to demonstrate the impact of both processes for conduction in the MOF, we calculated the electronic coupling between dimers of the linkers located in the neighboring layer (i.e. direct coupling, Figure 5a) and between trimers (coupling including superexchange-like process, Figure 5b) as depicted in Figure 4b.

Frequently observed high coupling values in Figure 5b mean that the "mediating" linker from another axis (see structures marked in orange circle in Figure 4b) fills pi-pi interactions with linkers that are on the consecutive layers and takes the position in between these linkers to maximize the attractive interaction. Since MC linkers are rather flexible and mobile at 300K, this "mediating" linker changes its location, so the electronic coupling fluctuates. Higher values of the coupling in Figure 5b indicate structures, where "mediating" linker is in between the consecutive linkers, where the superexchange-like coupling is the most favorable. The following text is now added in the revised manuscript:

"Frequently observed high coupling values in Figure 5b indicate that the mediating linker (from another axis than the consecutive linkers considered in the direct coupling, see structures in Figure 4b) fills $\pi \cdots \pi$ interactions with linkers and takes the position in between them to maximize the attractive interaction. The possibility for charge hops in this case increases, improving the conduction processes in the MC-based MOFs."

- In line 519, the authors state that they find a "four orders of magnitude improved conduction". As the conductivity is no longer explicitly calculated, this should be reworded.

Answer:

Thank you for pointing this out. We have corrected our statement in the Conclusions to:

"We find on average more than 23 and 65 times improved the electronic coupling between the HOMO and LUMO orbitals of the linkers in such MOFs upon spiropyran-to-merocyanine isomerization. This indicates, in approximation, more than 530 and 4200 times increase in the CT rate during on-off conduction photoswitching in the spiropyran-based MOFs, which depends quadratically on the electronic coupling."

minor:

- I think it would help to more clearly discuss the limitations of the major assumptions of a 100% switching rate and a misalignment of the linkers.

Answer:

We have added these statements to the revised paper:

"In the present case, we assume (ideal) 100% conversion between the photo-isomers, in order to understand possible on-off conduction ratios as the function of the intersheet distances and of chemical modifications of the linkers in the simulated MOF models."

"We exclude such effects in the present study, therefore, the calculated on-off photoswitching ratio in the present study should be considered as the highest possible value for the type of the MOF investigated."

"Since we considered 100% photoconversion of linkers in the pillared-layer MOFs, defect-free material and the starting location of the side chains of the layer linkers in the MOF as

pointing in the same direction, the on-off ratios reported should be treated as the maximal possible values that could be achieved for the photoswitching in spiropyran-based MOFs.”

- The authors might mention in the context of equation (6) that k_{CT} between neighboring linkers varies by orders of magnitude over time and that it is far from straightforward to determine an effective k_{CT} that one could directly insert into the equation. Still their time-resolved calculation of k_{CT} reveals clear trends.

Answer:

We have added this explanation in the description of results as:

“Due to the significant fluctuations of the electronic coupling over time at 300 K, there is no straightforward way to calculate the effective CT rate (eq. 1) using the average value of the coupling.”

- The entire discussion on page 11 is based on data that are only shown in the Supporting Information. Thus, it is very difficult to follow that discussion. I wonder, whether including at least the most relevant of those data also into the main manuscript would be beneficial.

Answer:

Thank for your suggestion. We agree with the reviewer. We have added two new Figures (Figure 6 in the main body and Figure S14 in the supplementary), where the comparison of the electronic couplings between HOMO and LUMO orbitals upon chemical modifications of SP and MC is depicted. It fits the description of results given in page 11 and simplifies data analysis.

- The discussion in lines 477-480 has been added in response to a comment of an issue with the original paper mentioned in my first report. As such, it has been a useful statement in the referee reply. I am, however, not sure, whether including it in its current form into the main manuscript is really useful, as there it is quite out of context and would probably just confuse the reader.

Answer:

Thank you. We have deleted the sentence in lines 477-480.

- sometimes the construction of the sentences is very “involved”.

Answer:

We have rephrased some sentences to make the context clearer. All changes are marked in yellow.

REVIEWERS' COMMENTS:

Reviewer #3 (Remarks to the Author):

The authors have all points addressed in the second repored and improved the manuscript. To me it is now suitable for publication.